# High spatial resolution retrieval of cloud droplet size distribution from polarized observations of the cloudbow

Veronika Pörtge[1], Tobias Kölling[1,2], Anna Weber[1], Lea Volkmer[1], Claudia Emde[1], Tobias Zinner[1], Linda Forster[1,3], and Bernhard Mayer[1]

[1]Meteorologisches Institut, Ludwig-Maximilians-Universität München, Munich, Germany
[2]now at Max Planck Institute for Meteorology, Hamburg, Germany
[3]now at Jet Propulsion Laboratory, California Institute of Technology, Pasadena, California

**Correspondence:** Veronika Pörtge (veronika.poertge@physik.uni-muenchen.de)

**Abstract.**

The cloud droplet size distribution is often described by a gamma distribution defined by the effective radius and the effective variance. The effective radius is directly related to the cloud's optical thickness which influences the radiative properties of a cloud. The effective variance affects, among other things, the evolution of precipitation. Both parameters can be retrieved from measurements of the cloudbow. The cloudbow or rainbow is an optical phenomenon, which forms by single scattering of radiation by liquid cloud droplets at the cloud edge. The polarized radiance of the cloudbow crucially depends on the cloud droplet size distribution. The effective radius and the effective variance can be retrieved by fitting model simulations (stored in a look-up table) to polarized cloudbow observations.

This study uses measurements from the wide-field polarization-sensitive camera of the LMU spectral imager system spec-MACS onboard the German research airplane HALO. Together with precise cloud geometry data derived by a stereographic method, a geolocalization of the observed clouds is possible. Observations of the same cloud from consecutive images are combined into one radiance measurement from multiple angles. Two case studies of trade wind cumulus clouds measured during the EUREC[4]A field campaign are presented and the cloudbow technique is demonstrated. The results are combined into maps of the effective radius and the effective variance with a spatial resolution of $100\,\text{m}$ by $100\,\text{m}$ and large coverage (across-track swath width: $8\,\text{km}$). The first case study shows a stratiform cloud deck with distinct patches of large effective radii up to $40\,\mu\text{m}$ and a median effective variance of $0.11$. specMACS measures at a very high angular resolution (binned to $0.3°$) which is necessary when large droplets are present. The second case study consists of small cumulus clouds (diameters of approximately $2\,\text{km}$). The retrieved effective radius is $7.0\,\mu\text{m}$ and the effective variance is $0.08$ (both median values). This study demonstrates that specMACS is able to determine the droplet size distribution of liquid water clouds even for small cumulus clouds which are a problem for traditional droplet size retrievals based on total reflectances.

## 1   Introduction

Clouds have two major implications on Earth's climate system. They contribute to the surface energy budget through latent heat release and directly interact with solar and terrestrial radiation. In addition, clouds can produce precipitation that strongly

affects our lives, especially in the case of extreme precipitation, which is characterized by its very high magnitude and its very rare occurrence at a specific location (IPCC, 2021). Clouds are complex phenomena, and understanding them is a challenging research topic. They can form in almost any region of the Earth and appear at different heights in the atmosphere. What makes them so complicated is, e.g., their high variability both in space and time. In addition, cloud particles have complex microphysical properties and exist in different thermodynamic phases (liquid, ice, supercooled liquid). This significantly impacts the radiative properties of a cloud. The study of clouds becomes even more difficult since aerosols must also be considered to better understand clouds. Aerosols serve as cloud condensation nuclei and affect clouds directly by changing the cloud droplet number concentration, but also by, e.g., suppressing rain, which in turn can change the cloud lifetime (Albrecht, 1989). Simulating clouds in models is challenging, not only because of the issues mentioned above but also because clouds occur at different scales. Their size can be as small as a few meters or as large as hundreds of kilometers, which is an issue for models as they are always limited by their resolution. Although there are cloud-resolving models, that substantially help in understanding clouds, such models are computationally very expensive and still rely on parameterizations which are subject to uncertainties (Satoh et al., 2019). At the same time, *measuring* clouds is equally difficult. In situ measurements accurately represent the atmospheric state of a few cubic centimeters, but this may not be representative for the cloud as a whole. Observing clouds by remote sensing instruments suffers from retrieval uncertainties and in general, improving models based on observations is not a straightforward task. Although the understanding of clouds has improved due to more and better observations as well as new cloud modeling approaches, the influence of clouds remains a large uncertainty in predicting future climate (Forster et al., 2021). This is why there is a great interest in extending our knowledge of clouds.

There are several past and planned field campaigns that aim at better understanding clouds and cloud feedback mechanisms (e.g., Arctic Cloud Observations Using Airborne Measurements during Polar Day (ACLOUD) and Physical Feedbacks of Arctic Boundary Layer, Sea Ice, Cloud and Aerosol (PASCAL), both presented in Wendisch et al. (2019), or the Next-Generation Aircraft Remote Sensing for Validation studies (NARVAL), see Stevens et al. (2019)). One such field campaign that put enormous effort into understanding clouds was EUREC[4]A (ElUcidating the RolE of Cloud-Circulation Coupling in ClimAte). The campaign took place in January and February 2020 and had its base in Barbados (Bony et al., 2017; Stevens et al., 2021). The goal was to intensely measure trade-wind clouds, which are the most frequent cloud type on Earth and therefore crucial for Earth's radiation budget. These clouds and how they react to climate change are a major source of uncertainty in climate sensitivity across different climate models (Bony and Dufresne, 2005). One of the many measurement platforms involved was the German research airplane HALO which was configured as a cloud-observatory similar to the previous NARVAL-II HALO campaign with radar, radiometer, lidar, and different spectral imagers (Stevens et al., 2019; Konow et al., 2021) including the cloud camera system specMACS (Ewald et al., 2016).

While EUREC[4]A studied clouds at many different scales, here we focus on observations of the microphysical properties of liquid water clouds. Two parameters, namely the effective droplet size and the width of the cloud droplet size distribution (DSD), are particularly important. The effective droplet size determines the radiative effect of clouds on the energy budget. A smaller droplet size (at a constant liquid water content) results in a large part of the incoming solar radiation being reflected by the cloud (Twomey, 1974). The width of the DSD influences the evolution of precipitation (Brenguier and Chaumat, 2001).

Often, the effective radius $r_{eff}$ is used as a quantitative description of the droplet size, and the width of the size distribution is characterized by the effective variance $v_{eff}$ (Hansen, 1971).

Cloud droplet size retrievals are often based on the bi-spectral technique which uses radiance measurements at two different wavelengths (Nakajima and King, 1990). Measurements at a wavelength in the visible wavelength region (VIS) such as $0.75\,\mu m$ where scattering dominates are combined with measurements at an absorbing wavelength in the shortwave-infrared (SWIR, e.g., $2.16\,\mu m$). This method simultaneously retrieves $r_{eff}$ and cloud optical thickness and is widely used for satellite instruments such as the Moderate Resolution Imaging Spectroradiometer (MODIS) (Platnick et al., 2003). The technique is well established, but it has known biases in the presence of 3-D effects and spatial inhomogeneity (Marshak et al., 2006; Zinner et al., 2008; Ewald et al., 2019). Furthermore, the bi-spectral technique does not provide information about $v_{eff}$ (Nakajima and King, 1990).

In recent years the use of polarized measurements for the retrieval of cloud (and aerosol) optical properties has become more and more popular (e.g., Bréon and Goloub (1998); Alexandrov et al. (2012a); Diner et al. (2013); Remer et al. (2019); McBride et al. (2020)). Polarized measurements have the advantage that multiple scattered contributions are filtered out and single scattering dominates the signal (Hansen, 1971). This greatly reduces 3-D effects which simplifies the analysis. Based on polarized observations of the cloudbow, a new type of DSD retrieval has emerged: the polarimetric technique. This method determines both the $r_{eff}$ and $v_{eff}$ of the DSD from polarized radiance measurements. The polarized radiance of liquid water clouds is sensitive to the $r_{eff}$ and the $v_{eff}$ in the region of the backscatter glory (scattering angle from $170°$ to $180°$), and in the cloudbow or rainbow region ($135°$ to $165°$). Both phenomena are described by Mie theory (Mie, 1908; Hansen, 1971). The polarimetric retrieval fits polarized phase functions against the measured polarized radiance (Bréon and Goloub, 1998) and will be discussed in more detail in Section 3.3. In general, unpolarized images also show the glory and the cloudbow and have already been successfully evaluated in terms of the DSD (e.g., Mayer et al., 2004). But especially for the cloudbow, the contrast in unpolarized observations is usually weak because the signal is dominated by the multiple scattering background. The use of polarized observations significantly enhances the signal.

One important aspect is to determine from which height within the cloud the measured signal originates. Here, the polarimetric retrieval has an advantage over the bi-spectral method since the bi-spectral signals come from a certain, not well defined, distance within the cloud as the photons are scattered multiple times until reaching the sensor (Platnick, 2000). The polarized signal, however, emerges from the cloud top within an optical depth of 1 (Alexandrov et al., 2012a), as the polarized signal is generated by singly-scattered photons. Knowing the location from where the signal emerges is required for the interpretation of the result. Furthermore, the $v_{eff}$ of the DSD is derived in the polarimetric retrieval. This parameter may be directly linked to entrainment and mixing processes at the cloud top.

The additional information from polarimetric measurements is also advantageous when it comes to studying aerosols (Remer et al., 2019). Aerosols and clouds have different angular polarimetric signatures (e.g., Emde et al., 2010), which can be exploited to distinguish between aerosols and clouds. Furthermore, theoretical studies showed that aerosol properties can be retrieved from polarimetric measurements with sufficient accuracy for climate research (e.g., Mishchenko and Travis (1997); Hasekamp and Landgraf (2007)). For instance, the simultaneous characterization of cloud properties and properties of aerosol above

clouds (Knobelspiesse et al., 2011), or of aerosol between clouds (Hasekamp, 2010; Stap et al., 2016a, b) is possible when using multi-angle polarimetric measurements. Obtaining polarization data from space is therefore desirable to improve the global picture of the atmosphere concerning both cloud and aerosol properties, and to quantify aerosol-cloud interactions. For this reason, several satellite missions with polarimetric instruments onboard will soon be launched or are already in space. The PACE (Plankton, Aerosol, Cloud, ocean Ecosystem) mission will be a polar-orbiting satellite that will deploy two polarimeters for cloud and ocean retrievals (Remer et al., 2019), the 3MI instrument (Multi-view Multi-spectral Multi-polarization Imager) will be part of the payload of the MetOp-SG satellite (Fougnie et al., 2018), and the MAIA instrument (Multi-Angle Imager for Aerosols) (Diner et al., 2018) will help to characterize particulate matter in air pollution, to name a few of the planned satellite instruments. The various existing polarimetric instruments, and those under development are listed in Dubovik et al. (2019). The development of polarimetric instruments is an active research focus and polarimetric airborne instruments are highly useful in investigating appropriate instrument design, satellite mission planning, or retrieval techniques.

As this work focuses on cloud measurements, we further want to highlight some instruments, to which the polarimetric cloud-bow retrieval has been applied successfully, such as POLDER (POLarization and Directionality of the Earth's Reflectances (Bréon and Goloub, 1998; Bréon and Doutriaux-Boucher, 2005; Shang et al., 2019)), RSP (Research Scanning Polarimeter (Cairns et al., 1999; Alexandrov et al., 2012a)), AirHARP (Airborne Hyper-Angular Rainbow Polarimeter (Martins et al., 2018; McBride et al., 2020)) or AirMSPI (Airborne Multi-angle SpectroPolarimetric Imager (Diner et al., 2013; Xu et al., 2018)). A detailed overview of instruments with polarization capabilities that also apply the polarimetric technique is given in McBride et al. (2020).

The retrieval technique has already been validated in several studies. For example, in 2013 the PODEX campaign took place (Knobelspiesse et al., 2019). This was an extensive intercomparison study between different polarimeters which, e.g., showed that RSP and AirMSPI measurements agree within the expected measurement uncertainties, especially for bright scenes (clouds, land). PODEX was carried out as preparation for the upcoming PACE mission. Alexandrov et al. (2018) compared in situ data to $r_{eff}$ and $v_{eff}$ results from the parametric fit of RSP measurements, and found a good agreement of better than $1\,\mu m$ for $r_{eff}$ and in most cases better than $0.02$ for $v_{eff}$. Painemal et al. (2021) compared the $r_{eff}$ and optical thickness of airborne data (polarimetric and bi-spectral retrieval based on RSP measurements and in situ measurements from the Cloud Droplet Probe) with satellite retrievals (MODIS and GOES-13) over the midlatitude North Atlantic. The comparison showed good correlations for the $r_{eff}$, but the satellite-based results were systematically higher than the aircraft measurements and the bias was larger for GOES-13 ($5.3\,\mu m$) than for MODIS ($2.6\,\mu m$). Recently, another comparison study was published by Fu et al. (2022), in which data collected during the Cloud, Aerosol and Monsoon Processes Philippines Experiment (CAMP2Ex) in 2019 were analyzed. One goal of the field campaign was to comprehensively compare $r_{eff}$ retrievals of cumulus clouds from different platforms (MODIS, RSP and in situ). RSP data can provide a bi-spectral and a polarimetric $r_{eff}$ from the same cloud target, due to spectral coverage from VIS to SWIR and along-track, co-located multi-angle sampling. The study shows that the $r_{eff}$ from the RSP polarimetric ($9.6\,\mu m$), the in situ ($11.0\,\mu m$) and the bias-adjusted MODIS $r_{eff}$ (Fu et al., 2019) ($10.4\,\mu m$) are in good agreement, but much smaller than the bi-spectral $r_{eff}$ from MODIS ($17.2\,\mu m$) and RSP ($15.1\,\mu m$). For shallow clouds, these differences are primarily caused by 3-D radiative transfer and cloud heterogeneity. There are several other studies, such

as by Bréon and Doutriaux-Boucher (2005), Di Noia et al. (2019) or Alexandrov et al. (2015) that compared $r_{eff}$ obtained from
polarized measurements with bi-spectral results and found similar biases. The differences could largely be attributed to the
different penetration depths of the SWIR band compared to the polarized signal, to differences in retrieval resolution, and to
3-D radiative transfer effects.

Here, we introduce the polarization upgrade of the airborne camera system specMACS (Ewald et al., 2016) and apply the
polarimetric technique to the specMACS measurements. In Section 2 we present the new polarization cameras of specMACS
in detail. Compared to other, already established polarimeters like RSP or AirMSPI, which operate in a scanning or pushbroom
mode, the specMACS polarization cameras capture a complete 2-D image of the observed scene at a high spatial resolution. The
special design of the Sony polarization sensor allows the simultaneous measurement of four different polarization directions
and three RGB color channels. The acquired images have a large field-of-view, which provides frequent observations of the
cloudbow. The polarimetric retrieval developed for deriving the DSD of liquid water clouds is discussed in Section 3 and
is applied to specMACS data that were measured during the EUREC[4]A field campaign in Section 4 where we present two
case studies. The first one is a stratiform cloud with two cloud layers at different heights. The second case study shows small
cumulus clouds (diameters $1\,km$ to $2\,km$). The results are presented as 2-D maps illustrating the high spatial resolution of the
specMACS measurements. Section 5 summarizes the results, compares the specMACS instrument to RSP and AirHARP, and
gives an outlook on planned future work with the specMACS data.

## 2   specMACS Polarization Cameras and Data Processing

The spectrometer of the Munich Aerosol Cloud Scanner (specMACS, Ewald et al., 2016) originally consisted of two hyper-
spectral line cameras sensitive in the wavelength range from $400\,nm$ to $2500\,nm$. During EUREC[4]A this set of cameras was
for the first time complemented by two identical polarization-sensitive imaging cameras. All four cameras are built into a pres-
surized, temperature stabilized, and humidity-controlled housing with a window in front of the cameras. The whole camera
system was flown in a nadir looking perspective onboard the German research airplane HALO (Krautstrunk and Giez, 2012).
In the past, the hyperspectral cameras have been successfully used to derive cloud droplet radius profiles (Ewald et al., 2019;
Polonik et al., 2020) or to retrieve cloud geometry from oxygen-A-band observations (Zinner et al., 2019). In this work, the
focus will be on the new polarization cameras.

The polarimeters are Phoenix polarization RGB cameras (Phoenix 5.0 MP Polarization Model) which come with Sony's
IMX250MYR CMOS polarized sensors with 2448 pixels (along-track) $\times$ 2048 pixels (across-track) (LUCID Vision Labs Inc.,
2022b). They are accompanied by a Cinegon 1.8/4.8 lens by Schneider-Kreuznach. The aperture is set to 5.6. The two cameras
are installed in a partly overlapping perspective which results in a combined maximum field-of-view of about $\pm 45°$ (along-
track) $\times \pm 59°$ (across-track). This corresponds to a horizontal pixel size at the ground of $10\,m$ to $20\,m$ at a cruise altitude
of about $10\,km$. The cameras are synchronized and measure at an acquisition frequency of $8\,Hz$. Furthermore, an automatic
exposure control system based on the method described in Ewald et al. (2016) is used to adjust the measurements to varying
illuminations.

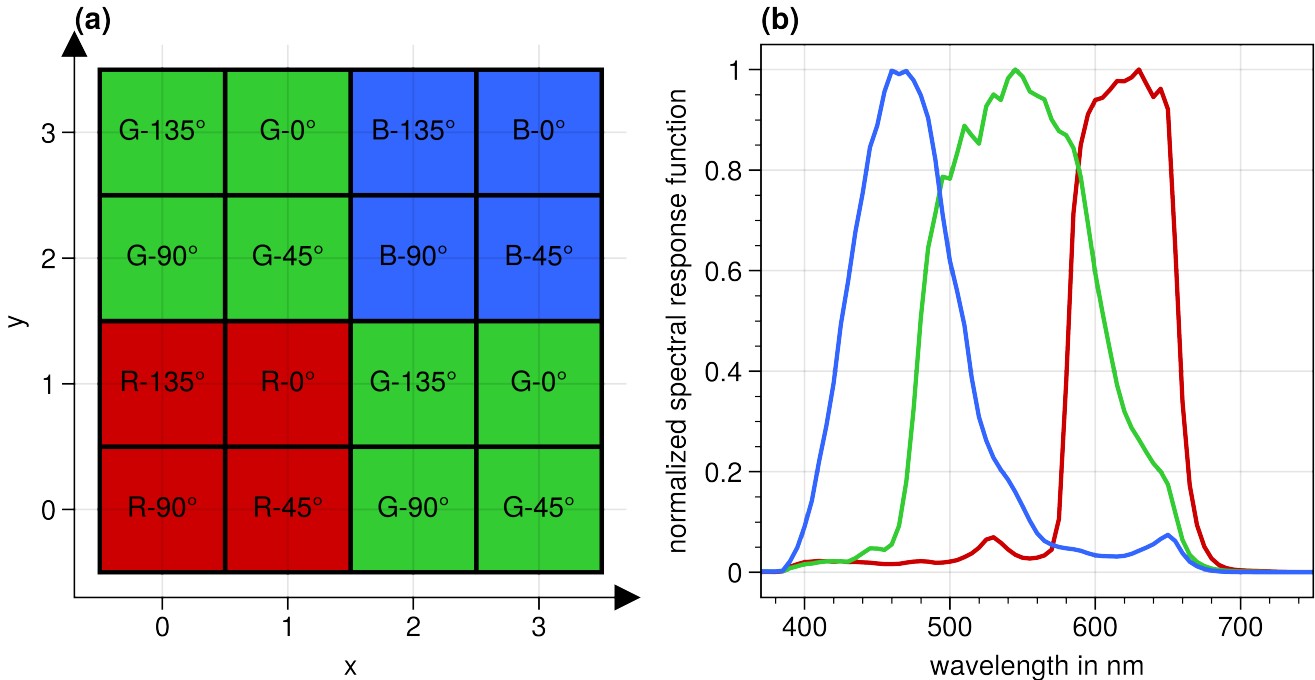

**Figure 1.** a) Structure of a 4 × 4 pixel block of the polarization cameras. Each 4 × 4 block is sub-divided into four blocks of 2 × 2 pixels for the different colors red (R), green (G), and blue (B). On the 2 × 2 pixel blocks four differently angled polarizers are placed. Figure adapted from the datasheet of the camera (LUCID Vision Labs Inc., 2022a). b) Normalized spectral response functions of the three color channels averaged over the four polarization directions taking into account the effect of the camera lens, and the window of the specMACS housing on the spectral response function.

The sensor accomplishes the measurement of polarization with on-chip directional polarizing filters (Fig. 1 a). The 2448 × 2048 pixels are split up into blocks of 4 × 4 adjacent pixels. These blocks are further divided into four 2 × 2 pixel blocks for each color of the color filter array (RGGB - Red, Green, Green, Blue). The spectral channels have center wavelengths (bandwidths) of approximately 620 nm (66 nm), 546 nm (117 nm), 468 nm (82 nm) (determined by a gaussian fit), and the normalized spectral response functions of each color channel are shown in Fig. 1 b). Polarizing filters (0°, 45°, 90°, 135°) are placed on top of each pixel (pixelated wire-grid polarizer). This enables the retrieval of three components ($I$, $Q$, $U$) of the Stokes vector of the light. The Stokes vector is a mathematical description of the polarization state of electromagnetic radiation and has four components (Hansen and Travis, 1974):

$$S = \begin{bmatrix} I \\ Q \\ U \\ V \end{bmatrix} = \begin{bmatrix} I_{0°} + I_{90°} \\ I_{0°} - I_{90°} \\ I_{45°} - I_{135°} \\ I_{\text{right-handed polarization}} - I_{\text{left-handed polarization}} \end{bmatrix} \tag{1}$$

$I$ is the total intensity and $Q$ and $U$ describe the linear polarization. The last component of the Stokes vector ($V$), which cannot be measured by specMACS, specifies the circular polarization. However, circular polarization does not play a role in cloud remote sensing since it is orders of magnitude smaller than linear polarization (e.g. Emde et al. (2015); Hansen and Travis (1974)). The degree of linear polarization (DOLP) describes the fraction of the incoming light that is linearly polarized

and is defined by $\text{DOLP} = \sqrt{Q^2 + U^2}/I$.

Figure 2 displays a specMACS measurement from the 2nd Feb 2020. The upper panels show the measurements of the *polA* camera which observes clouds slightly to the left in flight direction, the lower panel corresponds to the *polB* measurements slightly to the right in flight direction. On the left side, the measured total intensities of the two cameras are shown. Dashed lines indicate lines of constant scattering angle. The corresponding DOLP is shown on the right side of Fig. 2. Most parts of the

180 measurement have a small DOLP (dark in the image). The cloudbow region (scattering angle 135° to 165°) and the backscatter glory (scattering angle 170° to 180°) stand out due to their high DOLP. To avoid interpolation errors, we use the original data from the two individual cameras here, instead of projecting the data into a common mapping/figure.

A Stokes vector is defined with respect to a plane of reference. Often, the scattering plane, which contains both the incoming solar illumination vector and the view vector, is used as a reference plane (e.g., Eshelman and Shaw, 2019). This has the

185 advantage that $U \approx 0$ within the scattering plane and $Q$ contains all information about the polarized signal. In the case of the measurements, the original reference plane is the x-z-plane of the camera coordinate system. The x-axis of the camera coordinate system points into the flight direction which is also the polarizing axis of the 0° filter. The z-axis points in the direction of the optical axis of the camera. For further analysis, each measured Stokes vector is rotated into its pixel unique scattering plane (Hansen and Travis, 1974; Eshelman and Shaw, 2019) and we only evaluate $Q$. The window in front of the

190 polarization cameras affects the polarization state of the measurements. To correct for this effect, the window is handled as a linear diattenuator, and the Mueller matrix of a linear diattenuator is applied to the measurements (Bass et al., 1995).

A geometric calibration of the cameras was carried out using the chessboard calibration method described in Kölling et al. (2019) based on Zhang (2000), but we exchanged the thin prism camera model used in Kölling et al. (2019) by the rational model. Both camera models come from the OpenCV library (Bradski, 2000). In order to calculate the pixel coordinates of

195 specific 3-D points, the location and orientation of the camera with respect to a fixed world coordinate system have to be determined. The required precise information about the position and attitude of the aircraft is part of the Basic HALO Measurement and Sensor system (BAHAMAS) dataset. A high precision GNSS aided inertial reference system delivers the data with 100 Hz. The accuracy of the data is further increased by GNSS post processing after the flight (Giez et al., 2021). The camera location and orientation relative to the airframe is determined from the measured aircraft position and the location of distinct features

like rivers or roads in the images once after installation.

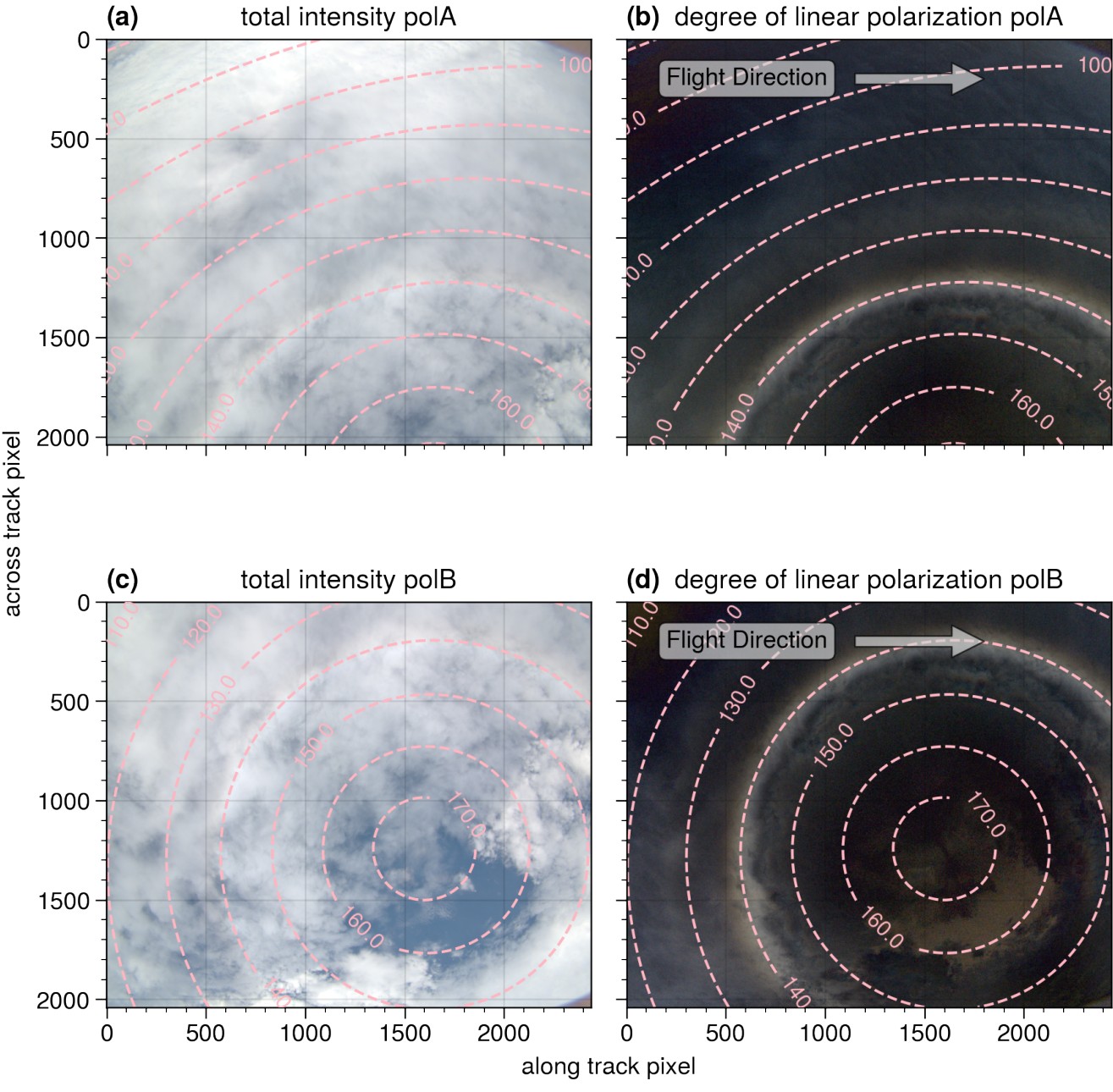

**Figure 2.** Example of measurements of both polarization cameras (2020-02-02 16:47:45.07 UTC). Top: Measurements from the first polarization camera (*polA*). This camera looks slightly to the left in flight direction. Bottom: Measurements from the second polarization camera (*polB*) which looks slightly to the right in flight direction. The field-of-view of the two cameras overlaps. Left: total intensity, right: DOLP. The dashed lines indicate lines of constant scattering angles in degree. The primary bow of the cloudbow is visible in the DOLP as a bright ring at a scattering angle of about 140°.

# 3 Retrieval Description

The goal of our algorithm is to determine the size distribution of cloud droplets from angularly resolved cloudbow measurements. An average cloudbow signal could be extracted from a cross-section of a single measurement (e.g., from Fig. 2). This method can easily be applied to any cloudbow observations, including those from commercial cameras, but the signal comes from a large area. The method presented in this paper is based on co-located observations along the track, which allows the acquisition of the cloudbow signature of individual targets. As a result, distributions are obtained at a high spatial resolution because this method does not involve averaging over a large area. With HALO, we fly over the clouds at a speed of about $200\,\mathrm{m\,s^{-1}}$, observing the same cloud from different viewing directions. Instead of evaluating the cloudbow in individual images, different viewing directions are sampled for each target on the cloud as specMACS images the scene (illustrated in Fig. 3). Similar approaches were also applied to measurements of other airborne and space-borne instruments (e.g., Bréon and Goloub (1998); Alexandrov et al. (2015); McBride et al. (2020)). The retrieval consists of three steps. First, cloud surface locations ("cloud targets") in the real world 3-D space and their trajectory caused by the wind are determined. For this purpose, we combine each $10 \times 10$ block of pixels from the specMACS images into target pixels. Such a target pixel typically has a size of about $100\,\mathrm{m} \times 100\,\mathrm{m}$ but the actual size depends on the distance to the cloud. We decided to use this target pixel size because it matches our pointing accuracy. Second, for each cloud target, the pixels of all images observing that location are collected. The individual measurements of one target are aggregated into a combined radiance measurement for the entire range of the viewing directions. In a final step, a look-up table (LUT) based on Mie calculations of polarized phase functions for different DSDs is fitted to the angular distributions to retrieve the best fitting DSDs. The particular steps of the aggregation process and the retrieval are described in the following.

## 3.1 Cloud Detection

The first step of the algorithm consists of detecting clouds in the measurements. As most measurements were taken above the ocean, the measurements are often contaminated with sunglint which appears due to the specular reflection of sunlight at the ocean. Cloud detection algorithms based on the brightness of the image often wrongly identify this bright sunglint as clouds. To (partially) overcome this problem we use the parallel component of the polarized light for the cloud detection. In the parallel component, the reflectance of the sunglint is significantly reduced. At the Brewster angle ($\theta_B \approx 53.1°$ for an air-water interface) reflected light is even completely perpendicularly polarized (Bass et al., 1995). In the case of a scene with medium cloud coverage, the algorithm chooses the red channel of the parallel component for further processing. For scenes with high cloud coverage, the normalized red ($r$) to blue ($b$) ratio ($\mathrm{nrbr} = (b_\parallel - r_\parallel)/(b_\parallel + r_\parallel)$) is calculated. Based on a brightness histogram of the selected data, a threshold value that distinguishes between cloudy and cloud-free pixels is determined with the method described in Otsu (1979).

A cloudy pixel is suitable for the cloudbow algorithm if it is observed within all scattering angles from $135°$ to $165°$ during the measurement sequence (for the choice of the range see, e.g., Alexandrov et al. (2012a); McBride et al. (2020)). This of course depends on the solar geometry and the camera's viewing direction. Therefore, the next step is to identify the cloud

targets that meet this criterion. In the case of Fig. 2, the upper part of the measurement cannot be used for the cloudbow retrieval as these clouds are not observed from the full scattering angle range needed while the aircraft is flying above the cloud. The flight direction is to the right as indicated by the arrow in Fig. 2 and the scattering angles are shown as dashed, circular lines.

## 3.2 Geolocalization of cloud targets

In order to identify the same target in different observations, we first use the geometric calibration of the camera to determine the viewing angle of the target (see Section 2). To fully localize the target we need to know the distance between the aircraft and the target ($\Delta z$ in Fig. 3). The altitude of the airplane and thus of the camera is measured by the BAHAMAS system. The cloud top height is derived using a stereographic reconstruction method which determines the cloud geometry from specMACS measurements. This was demonstrated for measurements of the previous 2D RGB camera in Kölling et al. (2019). The method identifies pixels with prominent features which are detected in the following images by a matching contrast. To correct for horizontal displacements of the cloud, the method was extended to include data of the horizontal wind from the ERA5 reanalysis dataset (Hersbach et al., 2020, 2018). The dataset has an hourly temporal resolution, a horizontal resolution of $0.25° \times 0.25°$, and 37 vertical levels from the surface to $1\,hPa$. During EUREC[4]A, clouds were typically observed at a vertical altitude of $1\,km$ to $2\,km$ where the ERA5 dataset has a vertical grid spacing of about $250\,m$. First, the stereo method is performed without additional wind information, and the 3-D coordinates of the identified pixels (stereo points) are retrieved. Then, the ERA5 data are interpolated to these coordinates to extract the corresponding wind data. The stereo method is performed again, but this time taking the wind data into account. The whole process is iteratively repeated 5 times, each time updating the wind data with the ERA5 wind interpolated to the heights and locations of the previously found stereo points. Further increasing the number of iterations did not notably change the results.

Figure 4 a) shows an example of the derived cloud top height of the *polB* camera using the stereographic method for the scene shown in Fig. 2 c) and d) (2020-02-02 16:47:45 UTC). Although the method has difficulties for homogeneous regions of the cloud due to a lack of contrast (e.g., in the lower right), large parts of the cloud are analyzed successfully. The cloud top heights from the single points of the stereographic method are interpolated to the entire image (Fig. 4 c). The interpolation process first consists of a linear interpolation of the stereo pixels onto all image pixels inside the convex hull of the stereo pixels. Then, the regions outside the convex hull of the original stereo points are filled by a nearest neighbor interpolation. The resulting cloud top heights are assigned to the selected cloud targets.

The WALES lidar system was also operated onboard HALO during EUREC[4]A (Wirth et al., 2009; Konow et al., 2021). The stereographically derived cloud top height is very similar to the measured cloud top height from the WALES lidar (Wirth, 2021) which is projected onto the specMACS RGB image in Fig. 4 b). Panel c) plots the WALES track on top of the interpolated specMACS cloud top height map. Within the high cloud on the left, the WALES data agree very well with the specMACS cloud top height and it is hard to distinguish the WALES data from the stereo data. The two datasets differ for the cloud on the right, where the stereo result is approximately $1000\,m$ lower than the lidar measurement. From the videos of the specMACS measurements, it can be seen, that here, the two cloud layers slightly overlap. specMACS detects the lower cloud layer due to

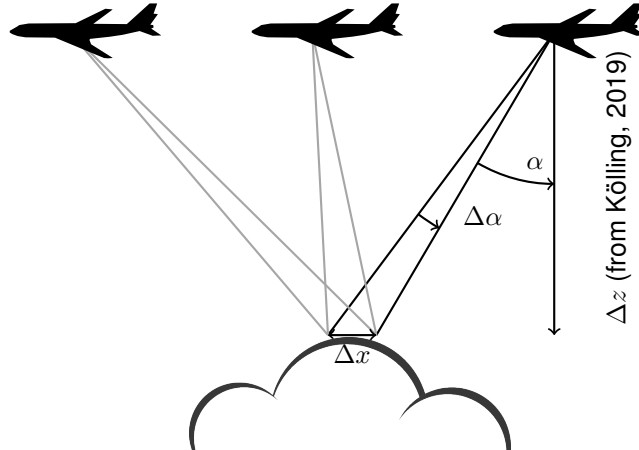

**Figure 3.** Observation geometry: The same target on the cloud (indicated by $\Delta x$) is observed from different viewing angles ($\alpha$). The cloud top height information needed to calculate the distance $\Delta z$ between target and camera is retrieved using the method described in Kölling et al. (2019). The single measurements are then aggregated into one radiance measurement of the target.

greater contrasts, while WALES is sensitive to the upper cloud layer. This behavior was also observed in Kölling et al. (2019). The stripe marked by the yellow lines in panel c) roughly surrounds the WALES track and defines the area for which the yellow
histogram in panel d) is derived. The cloud top heights of the two cloud layers are at approximately $2700\,\mathrm{m}$ and $1700\,\mathrm{m}$ (Fig. 4 d). The distribution of the interpolated stereo points is quite similar to the distribution of the WALES data (shown in blue), even though the two datasets differ for the cloud on the right.

Even a small error of a few hundred meters in the cloud top height will result in an erroneous localization of the cloud in subsequent images. An incorrect localization particularly affects targets close to cloud edges, where it will cause non-cloud
regions to be aggregated into the final cloudbow signal. Luckily, the stereographic method can very accurately determine the cloud geometry at cloud edges due to high contrasts.

By combining the cloud top height with the viewing directions, the locations of the cloud targets in the real world 3-D space are determined. These are used to calculate the pixel coordinates of the targets in successive measurements (Fig. 3), again considering the shift of the targets with the wind. The individual measurements of the same target of the Stokes parameter $Q$
are combined to generate the aggregated polarized radiance measurement. For further processing, the aggregated measurement is binned onto a scattering angle grid with a step size of $0.3°$. It should be noted that although specMACS has two polarization cameras with a partly overlapping field-of-view, until now, we do not combine the measurements of the two cameras. The results that are presented in the following are based on measurements of only one camera.

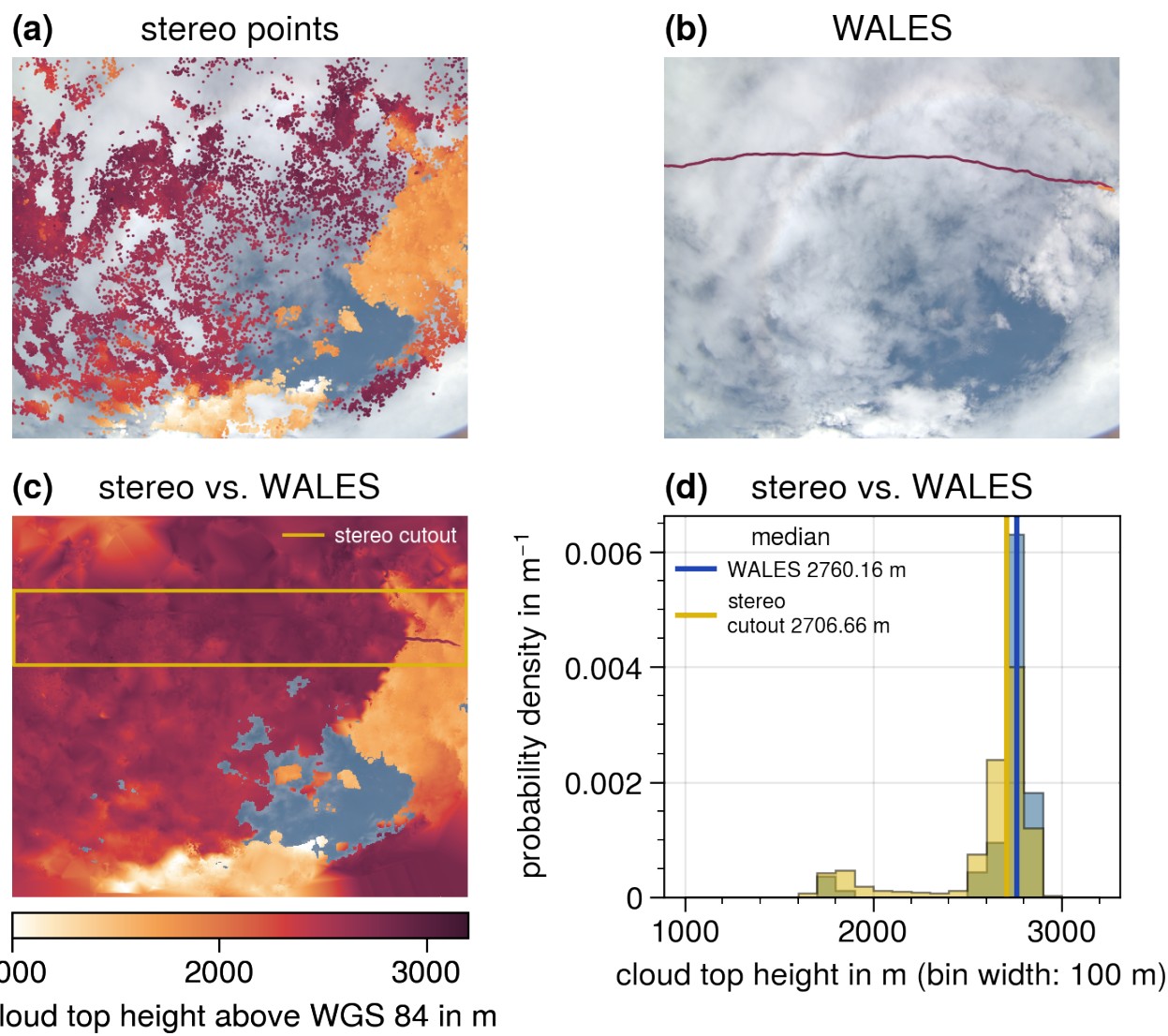

**Figure 4.** Cloud top height (CTH) information of the cloud field shown in Fig. 2 (2020-02-02 16:47:45.07 UTC). Panel a) CTH of stereo points from the stereographic reconstruction method; b) CTH from the WALES lidar system; c) Interpolated CTH based on the stereo points. The WALES CTH is plotted on top (hardly visible due to the similarity to the CTH from the stereo points). The stripe marked by the yellow lines indicates a specMACS cutout surrounding the WALES track. Panel d) shows the probability densities of the CTHs of the specMACS cutout (yellow) and the WALES measurements (blue). The RGB measurement of the cloud field is shown in the background of the panels a), b), and c). The colorbar below panel c) corresponds to all cloud top height measurements shown in the panels a), b) and c).

## 3.3 Size distribution retrieval

Polarized measurements are dominated by single scattering (Hansen, 1971). In general, any scattering process is described by the scattering matrix or phase matrix which relates incident to scattered radiation (Hansen and Travis, 1974). The scattering matrix is a $4 \times 4$ matrix with matrix elements $P_{ij}$. The $P_{12}$ element is also called the polarized phase function and is approximately proportional to the measured polarized radiance $Q$ in the scattering plane (Bréon and Goloub, 1998). Under the assumption of single scattering, it is true, that $P_{12}$ is directly proportional to $Q$ in the scattering plane.

Figure 5 shows examples of the polarized phase function for different $r_{\text{eff}}$ (a) and different $v_{\text{eff}}$ (b). This figure illustrates how the properties of the cloud droplets determine the shape and structure of the phase function, and thus the radiance within the cloudbow and glory region. The position of the maxima and minima of the polarized phase function strongly depends on the $r_{\text{eff}}$ (left figure). The $v_{\text{eff}}$, however, determines the amplitude and widths of secondary minima of the radiance distribution but has only a small effect on the position of the minima. Analysing the backscatter glory is an extremely accurate method to

retrieve $r_{\text{eff}}$ and $v_{\text{eff}}$ (Spinhirne and Nakajima, 1994; Mayer et al., 2004). But the glory requires a special observation geometry, and can therefore only be evaluated for a small fraction of the image area. The cloudbow however, covers a large area and thus is easier to observe, while still depending strongly on the size distribution.

For evaluating the aggregated angular radiance measurement with regard to the cloud droplet size properties (see Fig. 5), a LUT of polarized phase functions ($P_{12}$) for different $r_{\text{eff}}$ and different $v_{\text{eff}}$ was created for each of the three spectral color

channels of the camera. All calculations were carried out with the Mie Tool of the library for radiative transfer (libRadtran) (Wiscombe, 1980; Mayer and Kylling, 2005; Emde et al., 2010, 2016). We assume that the DSD has the shape of a monomodal gamma distribution. This is an extensively used assumption (Alexandrov et al., 2015), which is, e.g., confirmed by in situ measurements of liquid water DSDs (e.g., Miles et al., 2000). The $r_{\text{eff}}$ and the $v_{\text{eff}}$ of any DSD are defined as (Hansen, 1971):

1. Effective radius

$$r_{\text{eff}} = \frac{\int_0^\infty r \pi r^2 n(r) dr}{\int_0^\infty \pi r^2 n(r) dr} \tag{2}$$

2. Effective variance

$$v_{\text{eff}} = \frac{1}{r_{\text{eff}}^2} \frac{\int_0^\infty (r - r_{\text{eff}})^2 \pi r^2 n(r) dr}{\int_0^\infty \pi r^2 n(r) dr} \tag{3}$$

Here, $r$ is the droplet radius and $n(r)$ is the DSD. The formula of the gamma distribution can be written as a function of $r_{\text{eff}}$ and $v_{\text{eff}}$ (Hansen, 1971):

$$n_\gamma(r) = n_0 r^{(1-3v_{\text{eff}})/v_{\text{eff}}} \exp^{[-r/(r_{\text{eff}} v_{\text{eff}})]} \tag{4}$$

with:

$$n_0 = \frac{N(r_{\text{eff}} v_{\text{eff}})^{[(2v_{\text{eff}}-1)/v_{\text{eff}}]}}{\Gamma(\frac{1-2v_{\text{eff}}}{v_{\text{eff}}})}, \tag{5}$$

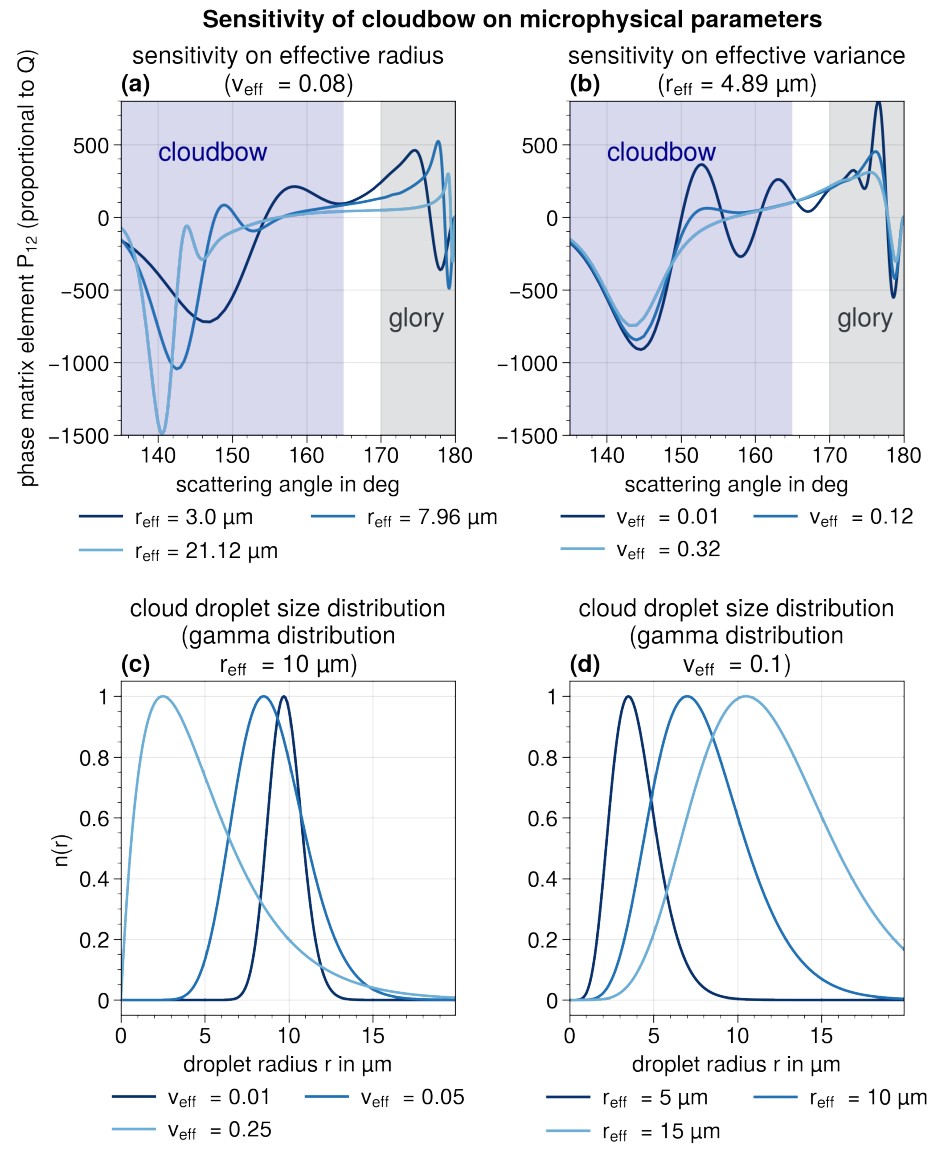

**Figure 5.** Plot a) shows that the cloudbow signals ($P_{12}$) vary if the $r_{eff}$ is changed while the $v_{eff}$ is constant ($v_{eff} = 0.08$). Plot b) illustrates the effect of a change in the $v_{eff}$ while the $r_{eff}$ is held constant ($r_{eff} = 4.89\,\mu m$). For calculating $P_{12}$, we assume, that the DSD has the shape of a gamma distribution. The $P_{12}$ curves shown here are for the green channel of the specMACS cameras. In the plots c) and d), several gamma distributions for different $v_{eff}$ and a constant $r_{eff} = 10\,\mu m$ (c) and for different $r_{eff}$ and a constant $v_{eff} = 0.1$ (d) are shown.

where $N$ is the total number of particles per unit volume. In Fig. 5 c) and d) several gamma distributions for different $v_{\text{eff}}$ (c) and $r_{\text{eff}}$ (d) are shown.

Polarized phase functions are calculated for a logarithmic grid of 77 different $r_{\text{eff}}$ ranging from $1\,\mu m$ to $40\,\mu m$ ($r_{\text{eff, i+1}} = r_{\text{eff, i}} \cdot 1.05$). The $v_{\text{eff}}$ range between $0.01$ and $0.325$ with a small step size of $0.01$ for $v_{\text{eff}} \leq 0.05$, and a larger step size ($0.02$ to $0.028$) for $v_{\text{eff}} > 0.05$. This choice is similar to other publications such as Alexandrov et al. (2012a); McBride et al. (2020). In total, the LUT includes 16 different $v_{\text{eff}}$. To account for the different spectral sensitivities of the three color channels, the polarized phase functions are initially calculated for the whole wavelength range of the spectral response functions with a step size of $10\,nm$, and are then weighted by each spectral response function (Fig. 1 b). For the calculation of the phase functions, a wavelength and temperature-dependent refractive index is used. We use the approximation formula of the IAPWS (International Association for the Properties of Water and Steam, Wagner and Pruß (2002)) for a temperature of $T = 10\,°C$ which, according to dropsondes measurements, corresponds to the approximate cloud top temperature of the typical EUREC[4]A clouds with a cloud top height of $1700\,m$.

The LUT of polarized phase functions ($P_{12}[r_{\text{eff}}, v_{\text{eff}}]$) is fitted to the aggregated radiance distributions ($Q_{\text{meas}}$) using the following equation:

$$Q_{fit}(\theta) = A \cdot P_{12}[r_{\text{eff}}, v_{\text{eff}}](\theta) + B \cdot \cos^2(\theta) + C \tag{6}$$

Here, $A$, $B$ and $C$ are fitting parameters and $\theta$ is the scattering angle. Parameter $A$ is needed to compare the radiometrically uncalibrated measurements with the simulated LUT, and, in addition, scales with the cloud fraction of the target made up of $10 \times 10$ pixels (Bréon and Goloub, 1998). The fitting parameters $B$ and $C$ account for any remaining effects that are not considered in the single scattering assumption. For example, these could be contributions by multiple scattering. The term $\cos^2(\theta)$ corrects for Rayleigh scattering contributions (Alexandrov et al., 2012a). Other studies do not rely on the cosine term, and instead use a correction term linear in $\theta$ plus a constant (e.g., Bréon and Goloub (1998); Bréon and Doutriaux-Boucher (2005)). In the cloudbow range, however, this is similar to $\cos^2(\theta)$ (Alexandrov et al., 2012a). A further contribution beyond single-scattering could be a thin cirrus cloud above the cloud that generates the cloudbow. In Riedi et al. (2010) it was shown that the polarization signal of ice particles depends linearly on the scattering angle in the rainbow region. Furthermore, Alexandrov et al. (2012a) showed that the magnitude of a cloudbow signal is attenuated by an overlying aerosol layer, but the aerosol layer does not change the structure of the cloudbow signal. The fitting parameters $B$ and $C$ also account for these two effects of cirrus and aerosols.

To determine $P_{12}$, and thereby the $r_{\text{eff}}$ and $v_{\text{eff}}$ of the DSD, a least-squares approach is used to invert Equation 6. In the inversion process, not only the grid points of the LUT are allowed, but also values in between. This is realized by a linear

interpolation of the LUT. The root mean square error (RMSE) is calculated for the scattering angle range from $135°$ to $165°$ where the cloudbow structure is most prominent:

$$\text{RMSE} = \sqrt{\frac{1}{n} \sum_{i=1}^{n} (Q_{\text{fit}}(\theta_i) - Q_{\text{meas}}(\theta_i))^2} \tag{7}$$

The smallest RMSE reveals the $r_{\text{eff}}$ and $v_{\text{eff}}$ of the DSD. In addition, the RMSE serves as a measure of accuracy and we filter out all fits with RMSE > 2.5. As a second quality measure, we calculate the quality index "Qual" as in Equation 8 (first defined by Bréon and Doutriaux-Boucher (2005)). This is the ratio between the variability of the measurement, which corresponds to the squared amplitude of the cloudbow ($A \cdot P_{12}$), and the RMSE of the fit. Measurements with a low quality index (Qual < 4) are filtered out of any further processing. This excludes, for example, "cloudbow signals" of ocean areas that have been incorrectly

identified as clouds from the result.

$$\text{Qual}^2 = \frac{A^2(\langle P_{12}^2 \rangle - \langle P_{12} \rangle^2)}{\text{RMSE}^2} \tag{8}$$

    Figure 6 shows two examples of aggregated cloudbow measurements for the green channel binned into $0.3°$ resolution in scattering angle (black dots with standard deviation and connecting black line). Each corresponding model fit is plotted as a solid, yellow line. The model fit matching example a) has $r_{\text{eff}} = 17.63 \, \mu\text{m}$ and $v_{\text{eff}} = 0.08$. Example b) has $r_{\text{eff}} = 5.98 \, \mu\text{m}$ and

$v_{\text{eff}} = 0.08$.

## 4   Retrieval Results

In the following, two case studies of the 2nd February 2020 are presented. On this day the observed clouds had a clear Flower organisation (Stevens et al., 2020; Dauhut et al., 2022). Such cloud fields are characterized by shallow trade-wind cumuli organized into large, stratiform clusters ($20 \, \text{km}$ to $200 \, \text{km}$) with high rain rates, surrounded by a large clear-sky region

(Stevens et al., 2020; Schulz et al., 2021). Among the four named mesoscale trade-wind cloud patterns (Flower, Gravel, Fish, Sugar), the Flower clouds have the highest cloud radiative effect (Bony et al., 2020), mostly because of their large low level cloud fraction. We demonstrate the polarimetric technique based on two case studies from specMACS measurements of this day. The first case study shows a part of the (stratiform) Flower structure. In the second example we analyze small trade-wind cumuli which were connected to a cold pool that formed during the dissipation of the Flower. We limit the presentation of the

retrieval results to the green channel, as the results from the red and blue channel are very similar.

### 4.1   Case Study 1 – Stratocumulus Flower Cloud System

First, we present the Flower cloud observed at 16:47:45 UTC. This measurement was already introduced in Section 3 and was shown in Fig. 2. At this time, HALO was flying at an altitude of about $10 \, \text{km}$ and the solar zenith angle was $31.15°$.

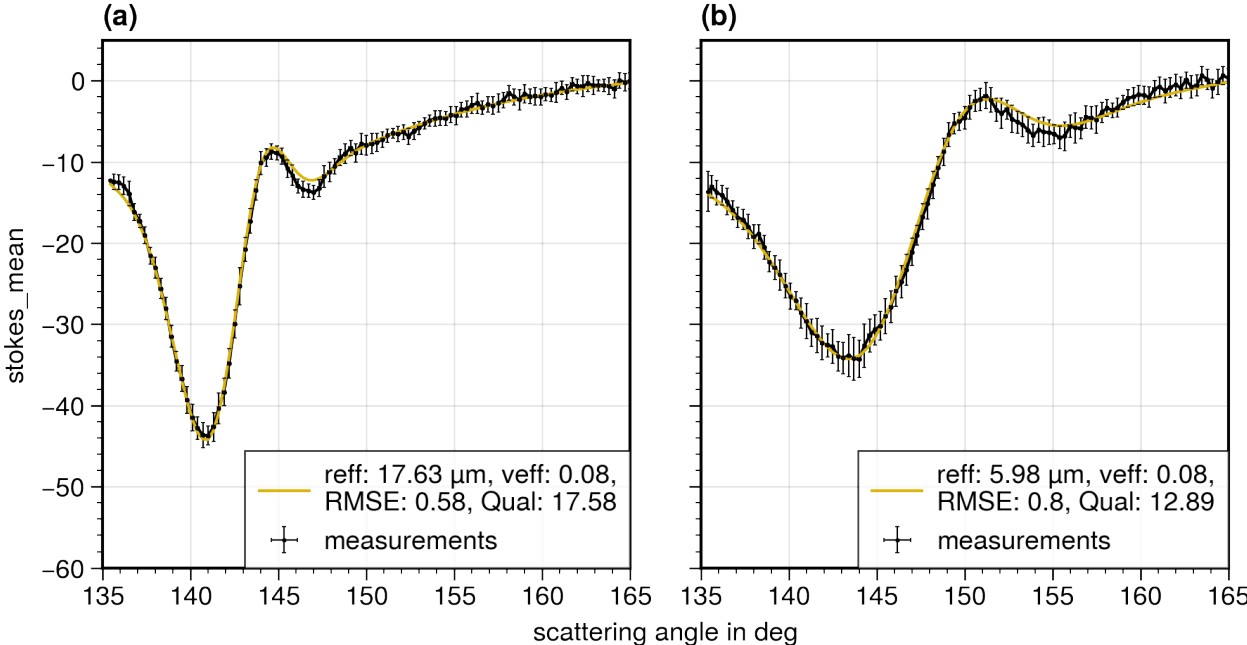

**Figure 6.** The two panels show the aggregated polarized radiance measurements of the green channel of two different target regions. The raw data are binned to $0.3°$ resolution in scattering angle (black dots connected by black lines). The error bars represent the standard deviations of all original data points within a $0.3°$ bin. The yellow lines indicate the best fitting simulations. The parameters $r_{eff}$, $v_{eff}$, RMSE, and Qual of the best fitting simulations are shown in the boxes in the lower right.

The cloudbow technique is applied to the measurements. The time required to sample the angular range $135°$ to $165°$ is $40\,s$. Fig. 7 g) shows the RGB image of the measurement from the *polB* camera. The labels on the four sides of the image indicate the distances between the neighboring corners of the image. It is noticeable that the side lengths of the top ($14.44\,km$) and bottom ($27.02\,km$) differ greatly. This happens, because the camera is installed at a slight angle in the across-track direction and therefore, the lower part of the image covers a much larger distance in the along-track direction. This is also the case for the measurements of the *polA* camera, but here, the upper part of the image covers a larger distance. The retrieval results of the individual cloud targets are combined into maps of $r_{eff}$ and $v_{eff}$ (Fig. 7 a and b). About one third of the image can be evaluated, as only the targets inside this area are observed from all necessary scattering angles during the overpass. The map of $r_{eff}$ (Fig. 7 a) is a consequence of the vertical distribution of the cloud field with two cloud layers at different cloud top heights (Fig. 4 and Fig. 7 h). The upper cloud deck at a height of about $2700\,m$ has a large $r_{eff}$ ranging between $15\,\mu m$ and $40\,\mu m$. Distinct patches of very large $r_{eff}$ values up to $40\,\mu m$ are observed. These patches occur in regions where the cloud is optically thick (Fig. 7 g). The spatial distribution of $r_{eff}$ of the lower cloud deck (cloud top height at $1700\,m$) is more homogeneous and the absolute values are much smaller ($r_{eff} \approx 6\,\mu m$). Figure 7 d) shows the frequency distribution of $r_{eff}$ where the two peaks of $r_{eff}$ of the two cloud decks are very well distinguishable.

The retrieved $r_{eff}$ values of the higher cloud are very large. To better understand the cloud field and the large $r_{eff}$ values, we looked at radar measurements of the polarimetric $K_a$-band MIRA-35 cloud radar of the HAMP instrument onboard HALO (Mech et al., 2014; Konow et al., 2021). The radar measurements from 16:47:00 to 16:48:30 UTC are shown in Fig. 8 along with a push-broom like image of the specMACS measurements and an indication of the HAMP radar field-of-view within the specMACS image. Within the high cloud from 16:47:00 to 16:48:15 the radar shows bands of enhanced reflectivity > 0 dBz and positive fall speeds (not shown). This likely corresponds to sedimenting droplets. Together with our observation of droplet sizes clearly larger than the usual cloud droplet size range (<15 μm) this points to drizzle development and we may see impacts of precipitation formation deeper in the cloud within the polarimetric signal originating from cloud top. Although our technique is not able to observe the precipitation droplet range (>100 μm) directly it is still sensitive to the intermediate size range below a possible drizzle droplet mode. This case study is particularly interesting, as the retrieved $r_{eff}$ lie within the size gap where neither the diffusional growth, nor growth by collision-coalescence is effective (Grabowski and Wang, 2013). A recent study by Sinclair et al. (2021) discussed the correlation between large cloud droplets detected by the RSP polarimeter and rain observed with a radar in great detail and found that the estimated cloud top precipitation rates are strongly correlated to radar derived precipitation rates and rainwater paths.

For our polarimetric technique, it is necessary to make an assumption on the shape of the DSD. Currently, we use a monomodal gamma distribution for this purpose. In Alexandrov et al. (2012b) it was shown that for clouds with a bimodal DSD (e.g., due to drizzle), the polarimetric retrieval based on monomodal DSDs is biased towards the dominant mode. To overcome this problem, the rainbow fourier transformation (RFT) was developed, that retrieves the DSD without any assumptions on the number of modes of the distribution (Alexandrov et al., 2012b). When comparing the polarimetric technique to the traditional bi-spectral retrieval, it should be noted, that bi-spectral retrievals are (normally) also based on simulations with monomodal DSDs (Platnick et al., 2017) which tends to underestimate the true $r_{eff}$ and has been investigated in several studies (e.g. Zinner et al., 2010; Zhang et al., 2012; Zhang, 2013).

The spatial distribution of $v_{eff}$ in Fig. 7 b) does not show a clear separation of the two cloud decks. Small patches of both very high and very small $v_{eff}$ can be seen. At the boundary between the two cloud decks, large $v_{eff}$ values are observed over several pixels. These are the result of a mixing of the signals of the two different cloud decks with different DSDs. Similar effects were seen in RSP observations of multi-layer clouds (Alexandrov et al., 2015, 2016). The resulting oscillating signal cannot be reproduced by a monomodal polarized phase function, and the outcoming fit has a large $v_{eff}$. The frequency distribution of $v_{eff}$ is shown in Fig. 7 e), and $v_{eff}$ has a median value of 0.11.

The panels c) and f) of Fig. 7 show the spatial distribution and the frequency distribution of the RMSE. The RMSE has a median value of 0.85 and there is no noticeable difference between the RMSE of the lower cloud with small $r_{eff}$ and the upper cloud with large $r_{eff}$. Small cracks are visible within the spatial distributions of $r_{eff}$, $v_{eff}$ and RMSE due to the reprojection of the targets onto the RGB image of the measurement, as there are small discontinuities within the interpolated stereo cloud top height. This in turn results in discontinuities within the locations of the reprojected targets.

The maps in Fig. 7 contain indicators of three particular cloud targets (colored circles). For these targets, the respective aggregated cloudbow measurements are plotted together with the model fits in Fig. 9. The targets a) and b) both lie within

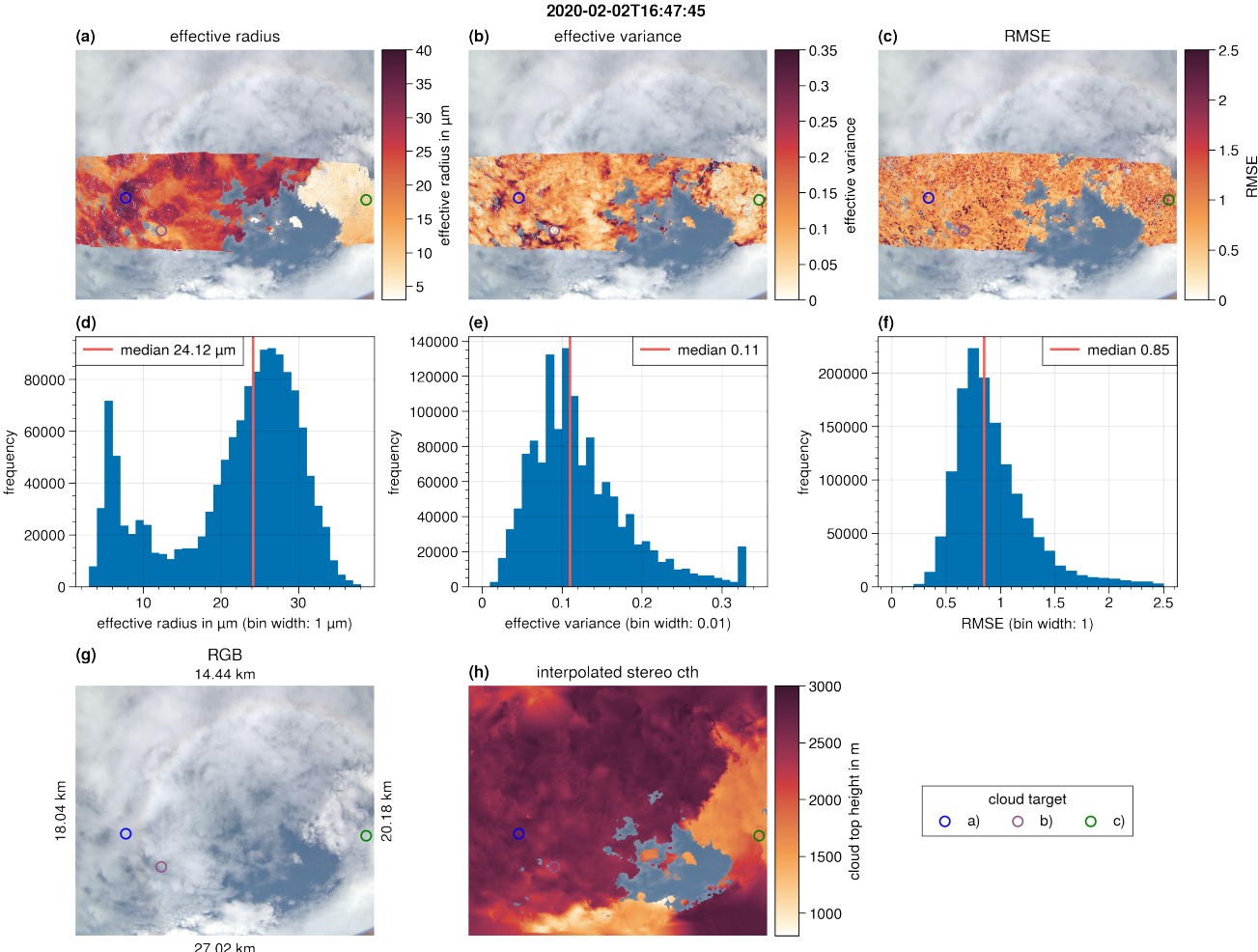

**Figure 7.** Spatial distributions of $r_{eff}$ (a), $v_{eff}$ (b) and RMSE (c) for the case study presented in Fig. 2. The panels d)-f) show the corresponding frequency distributions. Panel g) shows the RGB image of the measurement. The labels on the four sides of the RGB image indicate the distances between neighboring corners of the image. Panel h) shows the cloud top height from the stereo method interpolated onto the whole pixel grid of the image. Three specific cloud targets are indicated by colored circles in the maps.

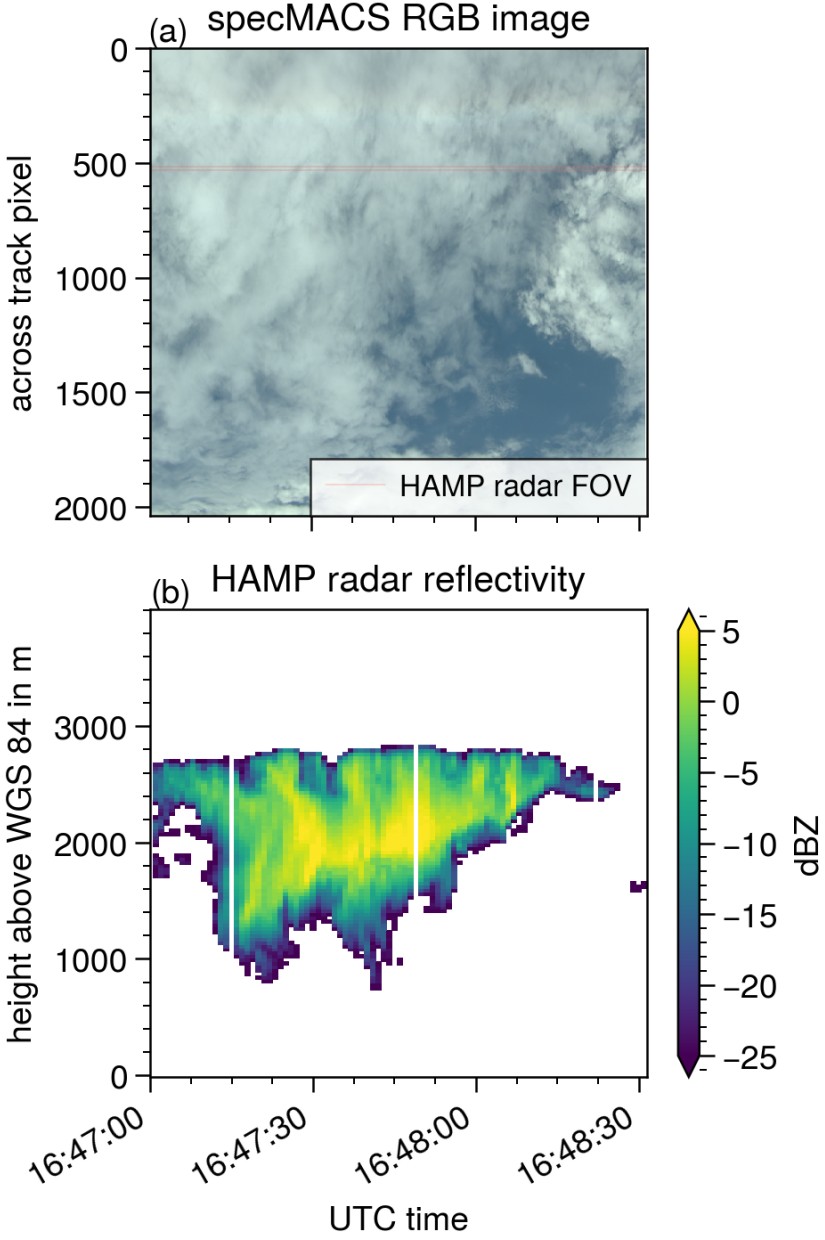

**Figure 8.** Temporal evolution of specMACS measurement (a) and HAMP radar reflectivity (b) for case study 1. The specMACS measurements are stacked together from individual images to generate a push-broom like image with a time axis. The HAMP radar field-of-view is marked within the specMACS image.

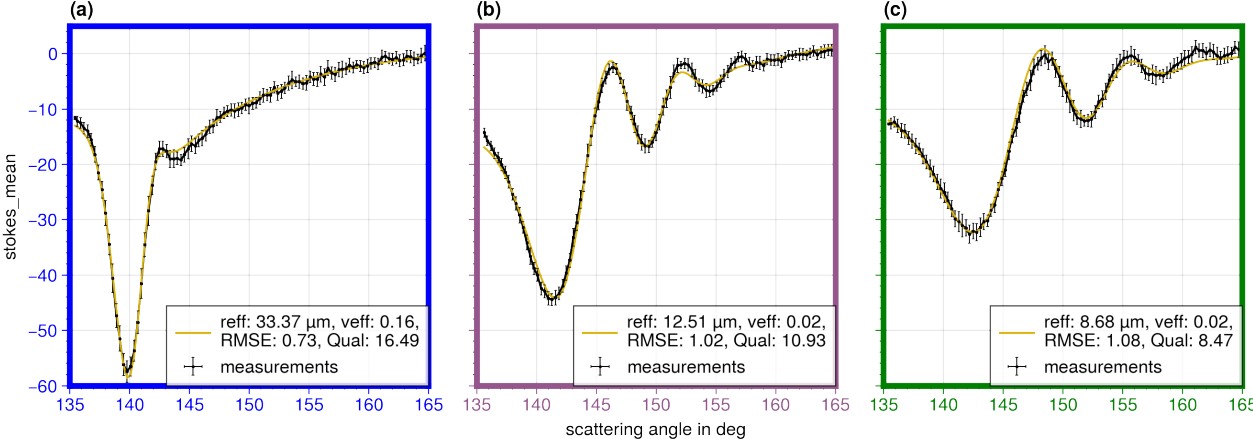

**Figure 9.** The aggregated polarized radiance measurements of the green channel of the locations shown in Fig. 7 were binned to $0.3°$ resolution in scattering angle (black dots connected by black lines). The error bars represent the standard deviations of all original data points within a $0.3°$ bin. The yellow lines indicate the best fitting simulations. The parameters $r_{eff}$, $v_{eff}$, RMSE, and Qual of the best fitting simulations are shown in the boxes in the lower right.

the high cloud deck. Target a) is located within a patch of very high $r_{eff}$. The corresponding cloudbow measurement has one very sharp minimum, and a second, weaker one. This indicates a relatively broad DSD. This is confirmed by the quite large

$v_{eff}$ value of 0.16. The measurement of target b) has several secondary minima. $v_{eff}$ is therefore reduced compared to target a) ($v_{eff}$ = 0.02). Target c) lies within the lower cloud deck. The cloudbow minimum is shifted to slightly larger scattering angles and the amplitude of the cloudbow is smaller than the amplitudes of targets a) and b). According to our expectations from the simulations (Fig. 5), this corresponds to a smaller $r_{eff}$, which is confirmed by the fit ($r_{eff}$ = 8.68 μm). The existence of the secondary minima indicates a narrow size distribution which is verified by the small $v_{eff}$ of the fit ($v_{eff}$ = 0.02). All three

measurements have only little noise indicated by the error bars.

## 4.2 Case Study 2 – Small Cumulus Clouds

In the following subsection, a second case study is discussed. The observations were taken from 18:28:15 UTC to 18:31:30 UTC with the polB camera. HALO was flying at an altitude of $10\,345\,m$ and the solar zenith angle was $46.1°$. The measurement shows a cloud field of small trade wind cumulus clouds with diameters of about $1\,km$ to $2\,km$ (Fig. 10 b). We are choosing

this example to demonstrate that the retrieval is capable of generating good results even in the case of more heterogeneous cloud scenes, and especially for small cumulus clouds. In such scenes, the traditional bi-spectral retrieval has issues with three dimensional radiative transfer effects (Marshak et al., 2006; Zhang et al., 2012). These are shadowing or illumination effects, which are normally not accounted for in standard radiance look-up tables.

In the case of small cumulus clouds, a precise geolocalization is important for image-to-image tracking. This geolocalization

depends on three factors: the internal calibration of the camera, the (mainly horizontal) wind at the cloud top, and the retrieved

cloud top height. In the following, we argue that, while these points certainly cause uncertainty in the geolocalization, they affect the cloudbow retrieval only by a lesser degree.

The internal calibration was verified using known targets at the ground. A slight deviation from the actual target of less than $100\,\mathrm{m}$ was observed, which also varied during the overflight. A possible reason for a slightly varying offset could be a degrading heading accuracy of the inertial reference system on long straight flight legs with no heading variation (Giez et al., 2021). As most of the EUREC[4]A flights were conducted on a circular flight pattern, the heading of the airplane continuously changed and the accuracy of the position and attitude data is very high. Since the inertial reference system is located in the nose of the airplane and the specMACS system in the tail, deformation of the fuselage caused by air turbulence or outside pressure change could also induce a varying offset in the pointing accuracy. Other inaccuracies result from the geometric chessboard calibration of the cameras and from the determination of the correct position of the specMACS instrument relative to the airframe by aligning specMACS measurements with satellite imagery. Based on this analysis of known ground targets, we used cloud targets of a size of approximately $100\,\mathrm{m}$ by $100\,\mathrm{m}$ in the current study. In the following, we will refer to this size of $100\,\mathrm{m}$ as "target unit". Future work will address improving the geometric calibration of the cameras to allow the study of even smaller targets.

The second factor that affects the geolocalization is the ambient horizontal wind. We apply a wind correction to the initial location of the cloud target to account for any drift due to the wind, and explain, why this is necessary in the following. According to the ERA5 data, the ambient horizontal wind of the case study at the cloud height (about $1\,\mathrm{km}$) was an east-southeast wind (direction $103°$) with a wind speed of about $6\,\mathrm{m\,s^{-1}}$. This is also confirmed by dropsondes measurements. During the flight, it took about $35\,\mathrm{s}$ to sample the targets from the angular range $135°$ to $165°$. During this aggregation process, a target cloud shifts by $210\,\mathrm{m}$ due to the wind. For a cloud target with a size of about $100\,\mathrm{m} \times 100\,\mathrm{m}$ this means that it moves further by more than two target units, therefore a wind correction is required. In addition, a cloud can evolve significantly within $35\,\mathrm{s}$, especially in the very active region at the cloud boundary, where the cloud grows or shrinks depending on cloud dynamics and the interaction with the environment.

The stereographic cloud geometry retrieval is very well applicable to this cloud field because of the strong contrasts between the clouds and the ocean. The resulting cloud top height (shown in Fig. 10 a) is relatively constant across the whole cloud field with a median value of about $1200\,\mathrm{m}$. Some (diameter wise) larger and more developed clouds also have higher cloud tops up to $2200\,\mathrm{m}$. Cloud top height data derived from WALES lidar measurements are projected onto the specMACS RGB image and are shown in Fig. 10 b). The lidar measurements are also plotted on top of the stereo points, which were interpolated onto the whole image (Fig. 10 c). The stereographic result is again similar to the WALES measurements. This is also evident in panel d), where the probability density of the stereo data in the surroundings of the WALES track (yellow rectangle in Panel c) is plotted along with the WALES cloud top height data (blue).

For a successful cloudbow retrieval, we rely on an accurate aggregation of the measurements by mapping from the known viewing angles to the image pixel location corresponding to the same cloud target. The stereographic approach of tracking cloud targets from one image frame to the next one provides exactly this information. It determines the cloud height by finding the ideal match between the known viewing directions during the overpass and the connecting line between identified targets

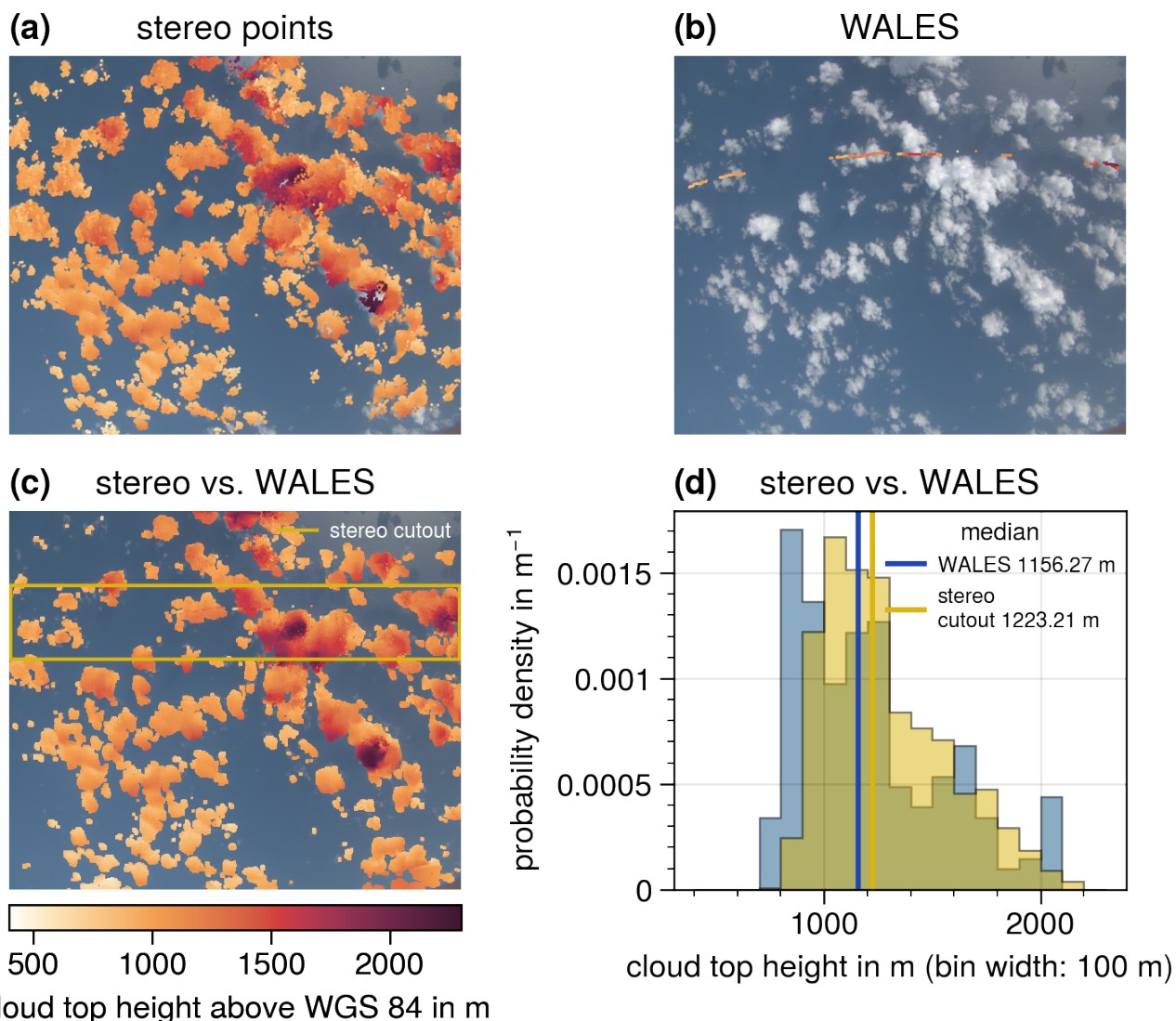

**Figure 10.** Cloud top height (CTH) data of the case study of small cumulus clouds. The measurement was taken on 2020-02-02 at 18:29:30 UTC. Panel a) CTH of stereo points from the stereographic reconstruction method; b) CTH from the WALES lidar system; c) Interpolated CTH based on the stereo points. The WALES CTH is plotted on top and the stripe marked by the yellow lines indicates a specMACS cutout surrounding the WALES track. Panel d) shows the probability densities of the CTHs of the specMACS cutout (yellow) and the WALES measurements (blue). The RGB measurement of the cloud field is shown in the background of the panels a), b), and c). The colorbar below panel c) corresponds to all cloud top height measurements shown in the panels a), b) and c).

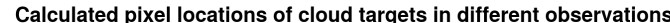

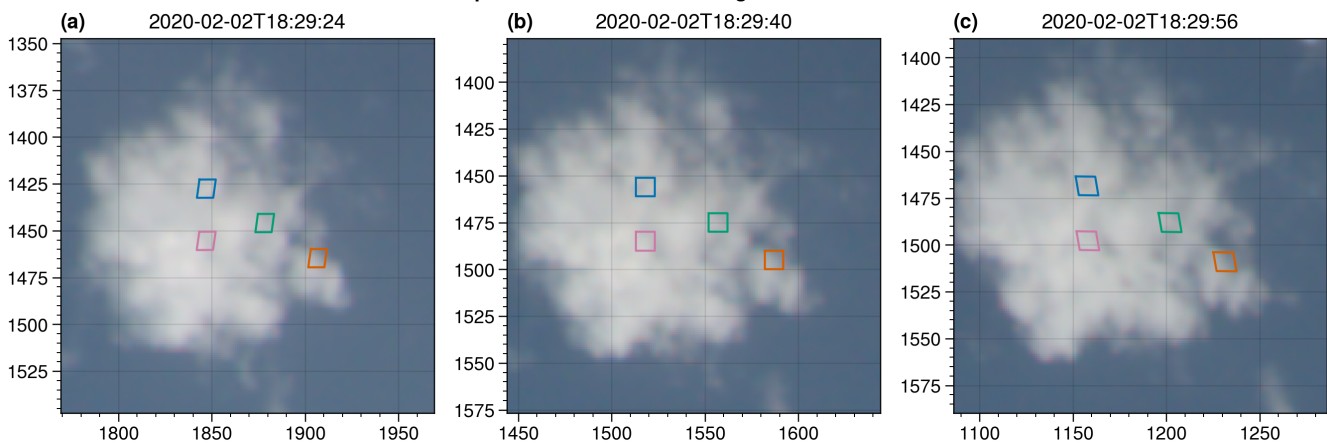

**Figure 11.** Calculated pixel positions of cloud targets, indicated by different colors, in observations at different times during the overflight.

at cloud top and the aircraft based on a matching contrast in different images. Some remaining residuum / mispointing error in the tracking, that stems from error sources in the geometric calibration (and as a result in cloud top height and horizontal wind), has negligible effects on the tracking. We manually verified the tracking of cloud targets with distinctive features during the overflight. One such example is shown in Fig. 11. Based on the location of the targets and the ambient wind at 18:29:40 UTC

(panel b), the pixel positions of the targets in a previous (panel a) and a later image (panel c) are calculated. A visual comparison of the identified targets in the different images shows that the targets are successfully tracked, as the colored markers in Fig. 11 highlight the same areas of the cloud in all three images. Due to camera distortions the shape of the originally rectangular cloud targets (at 18:29:40 UTC) increasingly takes the shape of a trapezoid when they approach the edge of the entire image. Each panel in Fig. 11 shows only a small part of the entire image.

The retrieved $r_{eff}$, $v_{eff}$ and RMSE results are projected onto the RGB image and shown in Fig. 12 a) - c). The corresponding frequency distributions are shown in the panels d) - f). The panels g) and h) show the RGB image of the measurement together with an indication of the dimension of the image and the interpolated cloud top height, respectively. Compared to case study 1, $r_{eff}$ is much smaller (median: $7.0\,\mu m$) and has a more narrow distribution. Values of $r_{eff}$ larger than $12\,\mu m$ are not observed. The spatial distribution of $r_{eff}$ is homogeneous and has only few outliers. For higher cloud tops, an increase in $r_{eff}$ is observed. This

dependence of the $r_{eff}$ on the cloud top height is presented in more detail in Figure 13. Here, the derived $r_{eff}$ of all individual clouds of the case study are plotted against the corresponding cloud top heights (as in Rosenfeld and Lensky (1998)). We refer to this plot as a vertical profile, although it does not show the actual $r_{eff}$ profile within a single cloud. The idea is that the individual clouds of the cloud field are captured at different stages of their vertical growth. It is then assumed that the retrieved $r_{eff}$ which is sampled at the cloud top is representative of the actual $r_{eff}$ at the same height inside a single cloud. This assumption

applies only to non-precipitating clouds. By combining the measurements of the individual clouds at different stages of their vertical development, a vertical profile is constructed, which is assumed to correspond to the vertical profile of a single growing

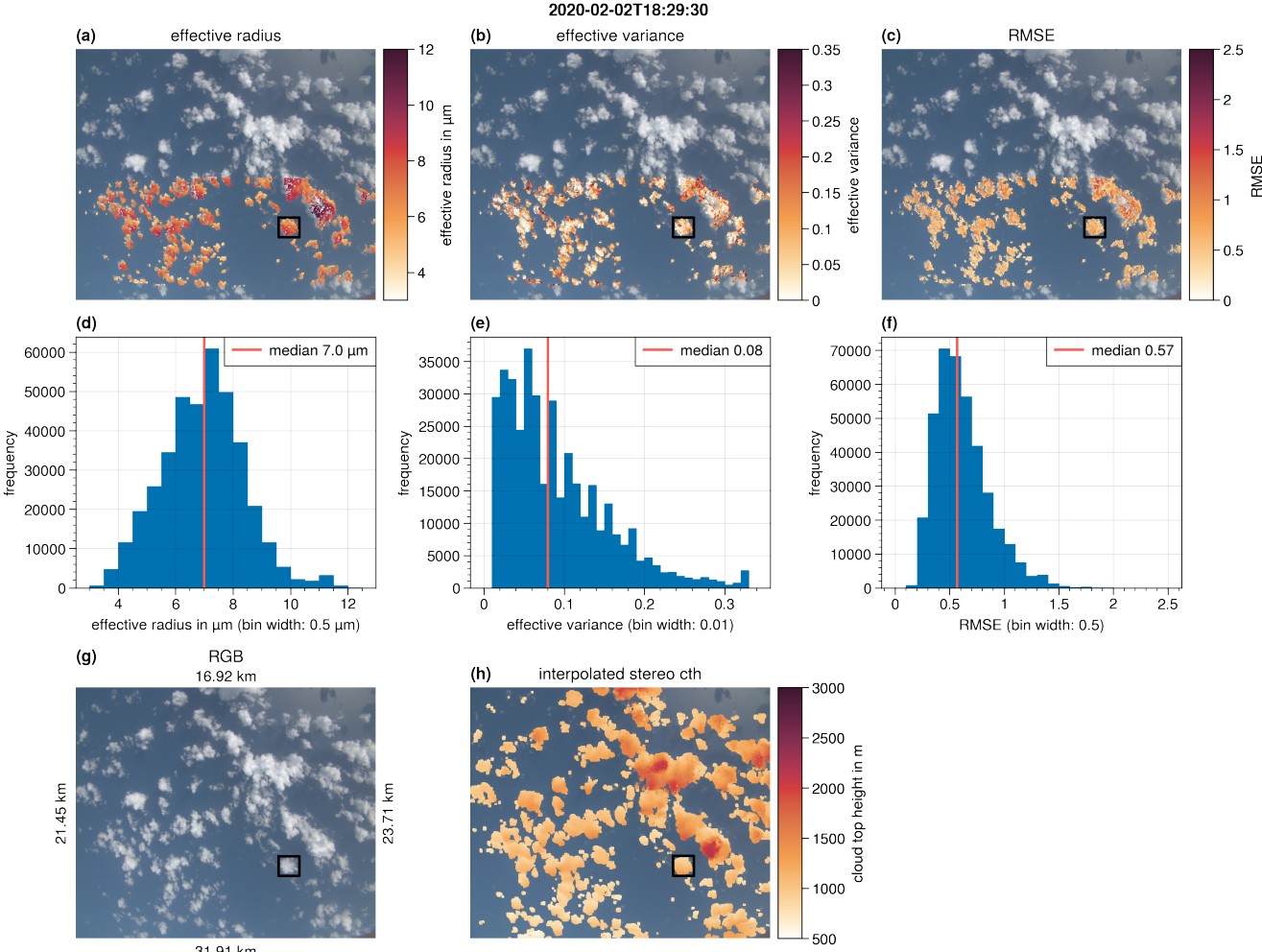

**Figure 12.** Same as in Fig. 7 but for case study 2 presented in Fig. 10. The black rectangle marks a single cloud, that is presented in Fig. 14.

cloud. The figure shows a strong increase of $r_{eff}$ from about $5\,\mu m$ at a height of $800\,m$ to $9\,\mu m$ at $1350\,m$. This rapid growth of droplets with height may indicate the dominance of growth through coalescence and is typical for maritime clouds. Rosenfeld and Lensky (1998) refer to this zone as the "droplet coalescence growth zone".

The $v_{eff}$ (Fig. 12 b and e) is small (median $v_{eff}$ = 0.08) and consistent within the inner part of the clouds. There are some cloud targets with $v_{eff}$ = 0.32 (the upper limit of the LUT), that occur mainly at the edge of the cloud. Especially at the edge of the cloud, a small offset in the geolocation can have a significant impact on the aggregated observations. The offset between the assumed location and the actual location may increase during the aggregation process and could even include ocean measurements for targets at the cloud edge. In this case, the aggregated measurements originate from different targets

and the cloudbow signal broadens or vanishes completely. We tried to ensure, that the RMSE and Qual criteria successfully

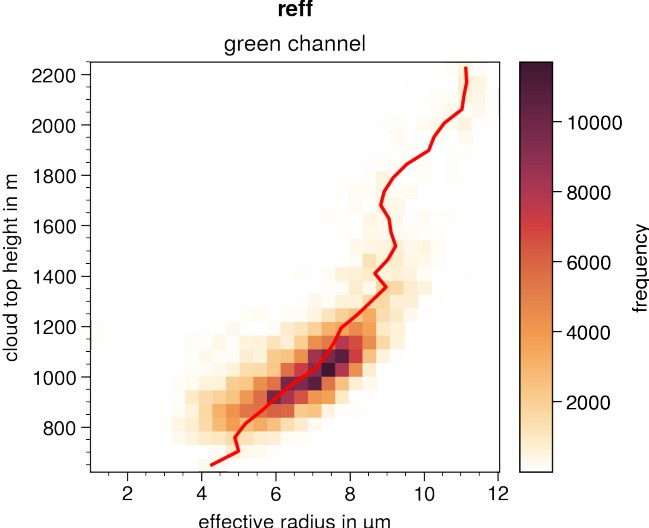

**Figure 13.** Vertical profile of the retrieved $r_{eff}$ of the cloud field shown in Fig. 12. The red line indicates the average $r_{eff}$ for each vertical layer.

filter out such cases. Furthermore, it could also be a physical effect related to entrainment and (inhomogeneous) mixing of dry air in the cloud. In this case, modeling studies predict a broadening (increase of $v_{eff}$) of the DSD (Pinsky et al., 2016). Such questions will be investigated in the future using the high-resolution specMACS data.

The black rectangle in Fig. 12 marks a single cloud (diameter $1.5\,\mathrm{km}$) that is presented in more detail in Fig. 14. In this figure, maps of the $r_{eff}$, the $v_{eff}$ and the cloud top height are shown together with the frequency distributions of $r_{eff}$ and $v_{eff}$ of the single cloud. The Qual and RMSE criteria filter out some of the cloud targets inside the cloud, which appear mainly in shadowed or optically thin parts of the cloud. The high spatial resolution of the measurements reveals small scale structures of the DSD, especially regarding the $v_{eff}$, that is, e.g., increased along a line from the top left to the center of the cloud. Three targets of the cloud are selected (marked by circles in the maps). The targets a) and b) have similar $r_{eff}$ but differ in the $v_{eff}$. Target c) lies within the region of increased cloud top height, where $r_{eff}$ is also large. The difference of these three targets is visible in the aggregated cloudbow observations presented in Fig. 15, which vary especially regarding the number and visibility of secondary minima. The observations are more noisy compared to the observations of case study 1 (Fig. 9), and also the absolute values of the cloudbow signals are less strong. This indicates, that even within one target the variability of the cloudbow signal is relatively large. A further reduction of the size of a target would be helpful, but this comes with the need of an even increased precision in geolocalization. Although the observations are more noisy, the primary cloudbow is still very pronounced which indicates that the retrieval of $r_{eff}$ is robust. Furthermore, $r_{eff}$ is for all three targets relatively small ($6.54\,\mathrm{\mu m}$ to $9.2\,\mathrm{\mu m}$). In this size range, the cloudbow signal depends strongly on $r_{eff}$ (see Fig. 5). The result of $v_{eff}$ is more difficult to interpret. The structure of the supernumerary bows (which mainly defines $v_{eff}$) can get smoothed out while averaging the signals of different DSDs within the averaging target and the resulting DSD is in the worst case different from any of the actual sub-pixel distributions. A

sensitivity analysis of the cloudbow algorithm based on different resolutions of AirHARP data was presented in McBride et al. (2020) to identify effects of sub-pixel variability. This analysis showed that in the case of a wide spread of the DSDs within the sub-pixels, the coarse resolution result may not reflect the mean of the sub-pixels, as the combination of different gamma size distributions from the sub-pixels is not another gamma size distribution (Shang et al., 2015). Furthermore, the AirHARP study used a single, constant cloud top height for the entire scene which introduces uncertainty in the retrieval (McBride et al., 2020).

In the future, the specMACS retrieval will be applied to even smaller targets which will further reduce effects of sub-pixel variability.

## 5  Discussion and Conclusion

We used the measurements of the new polarization resolving cameras of specMACS to retrieve the $r_{eff}$ and $v_{eff}$ of the DSD at the cloud top. The method relies on polarized measurements of the cloudbow which is sensitive to $r_{eff}$ and $v_{eff}$. Cloud top

height data from an existing stereographic retrieval (Kölling et al., 2019) are combined with the measured airplane position and attitude data to geolocate the measurements. A parametric fit is applied to all data points that contain the full cloudbow signature (scattering angle $135°$ to $165°$). The results of the cloudbow retrieval are combined into spatial maps of $r_{eff}$ and $v_{eff}$ that give new insights into cloud microphysics and the spatial distribution of the parameters at the cloud top. The maps reveal structures within the cloud that may be linked to dynamic processes. We presented two case studies of the EUREC$^4$A

campaign. The first study shows a stratiform cloud with two (mostly non-overlapping) cloud layers at different heights. In the higher cloud layer, large $r_{eff}$ ($25\,\mu m$ to $40\,\mu m$) are retrieved. These values correlate with bands of high radar reflectivity values indicating sedimenting droplets. The spatial maps are rather smooth and have only few outliers. The high spatial resolution of the retrieval results (currently about $100\,m \times 100\,m$) allows the observation of small cumulus clouds, that can be evaluated accurately using the polarimetric cloudbow technique. This is demonstrated in the second case study which shows cumulus

clouds with diameters of $1\,km$ to $2\,km$. The retrieved $r_{eff}$ values are much smaller ($3\,\mu m$ to $12\,\mu m$) and the $v_{eff}$ increases from the cloud center to the edge with a median value of $0.08$. During EUREC$^4$A many similar cloud fields were observed. Further evaluations of such cloud fields will include studying the effect of entrainment and mixing processes on the DSD of small cumulus clouds in more detail. Since the cloudbow is a single-scattering phenomenon, the results are less affected by 3-D radiative transfer effects in contrast to bi-spectral retrievals, which are not applicable to such small clouds.

In the past, similar methods were already applied to measurements of other instruments such as POLDER, RSP, or AirHARP. To situate specMACS in the scope of the already existing instruments, we summarize the main features of specMACS, and compare them to the technical details of the RSP and the AirHARP instruments. The main differences between the instruments are listed in Table 1 based on Alexandrov et al. (2012a) and McBride et al. (2020). The outcome of all three instruments' retrieval techniques are angularly resolved measurements of the Stokes parameters $I$, $Q$, and $U$. However, the way these

measurements are generated differs:

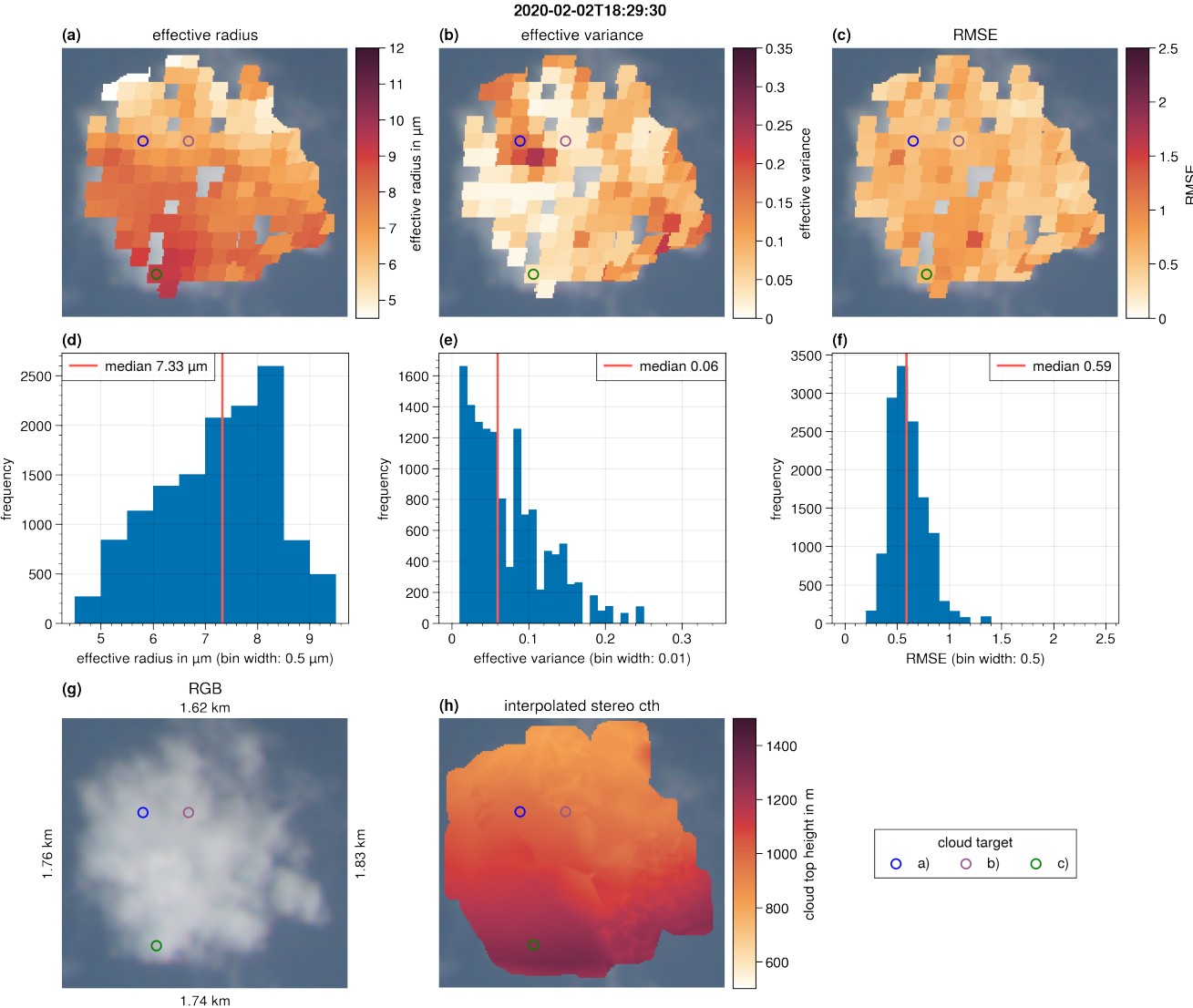

**Figure 14.** Same as in Fig. 7 but for zoom into one cloud shown in case study 2 (Fig. 10 and Fig. 12). Three specific cloud targets are indicated by colored circles in the maps.

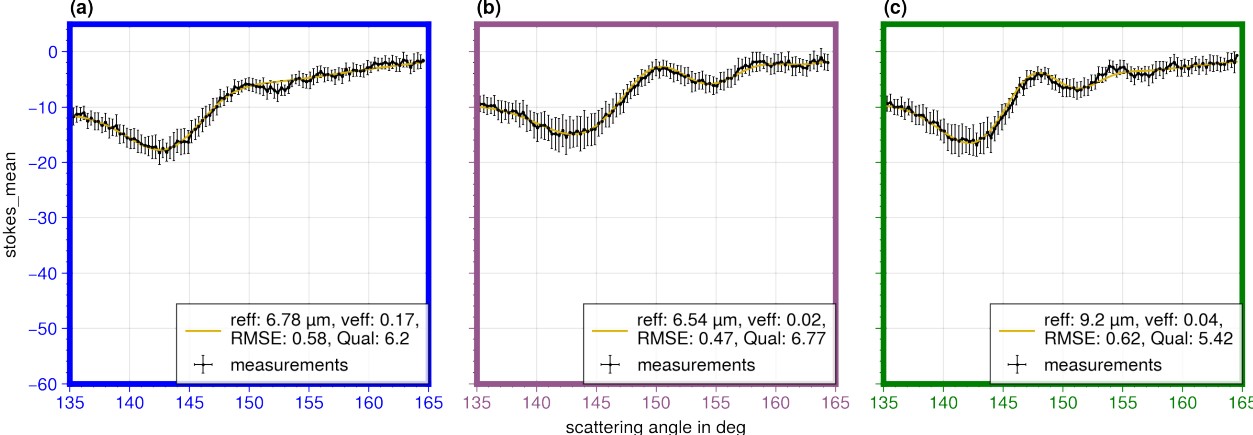

**Figure 15.** The aggregated polarized radiance measurements of the green channel of the locations shown in Fig. 14 were binned to $0.3°$ resolution in scattering angle (black dots connected by black lines). The error bars represent the standard deviations of all original data points within a $0.3°$ bin. The yellow lines indicate the best fitting simulations. The parameters $r_{eff}$, $v_{eff}$, RMSE, and Qual of the best fitting simulations are shown in the boxes in the lower right. For better visual comparison with the aggregated measurements of the first case study, the y-axis covers the same range as in Fig. 9.

- Each observation of the specMACS instrument is a 2-D image. Individual clouds are identified in successive images from different viewing directions and the subsequent single measurements are combined to generate angularly resolved cloudbow signals of each cloud.

- The AirHARP instrument is also an imaging instrument with a similar FOV as specMACS. There are 120 view sectors in the along-track direction, which all have a unique average viewing angle. The individual measurements of a single view sector are combined to generate a 2-D push broom image where all pixels are observed from the same viewing direction. Targets that are observed in multiple view sectors during the overflight can be used to generate angularly resolved reflectance measurements.

- RSP is an along-track scanner with only a single pixel in the across-track direction. During each RSP scan about 150 measurements are taken at $0.8°$ intervals. Data from all individual scans are aggregated into "virtual scans" which provide the full angular reflectance measurement at a single target. In addition to the common parametric fit retrieval, the RSP data can also be used to retrieve the DSD from the Rainbow Fourier Transform (RFT) technique which does not rely on an assumption on the number of modes of the DSD (Alexandrov et al., 2012b).

The major advantage of the specMACS and AirHARP instruments is their imaging capability with a large FOV. This not only increases the information content of the data but also makes it easier to measure the cloudbow since the cloudbow is observed within the FOV of the cameras for a large range of solar zenith angles. specMACS enables an even more detailed representation of the spatial distribution of the DSD, due to the higher spatial resolution $(100\,\mathrm{m})$ compared to AirHARP $(200\,\mathrm{m})$. RSP has the

**Table 1.** Technical details of specMACS, AirHARP (McBride et al., 2020), and RSP (Alexandrov et al., 2012a)

| | specMACS | AirHARP | RSP |
|---|---|---|---|
| field-of-view | maximum: $\pm 45° \times \pm 59°$ | $\pm 57° \times \pm 47°$ | $\pm 60°$ (along-track only) |
| spectral channels | three color channels (468, 546 and 620 nm) | four narrow spectral channels (440, 550, 670 and 870 nm) | nine narrow spectral channels (410, 470, 555, 670, 865, 960, 1590, 1880 and 2260 nm) |
| angular resolution | binned to 0.3° | 2° for 670 nm channel, all other channels: 6° | 0.8° |
| typical resolution of retrieval results (depends on distance to cloud) | approximately $100\,\text{m} \times 100\,\text{m}$ | $200\,\text{m} \times 200\,\text{m}$ | 120 m in Fu et al. (2022) |

highest number of spectral channels (9), including SWIR channels, and can therefore simultaneously retrieve the $r_{\text{eff}}$ based on the bi-spectral technique without any alignment errors. specMACS also offers the possibility for a bi-spectral retrieval because
of its additional two spectrometers, but these have a smaller FOV compared to the polarization cameras. Furthermore, RSP and AirHARP have narrower spectral channels compared to specMACS, which sharpens the cloudbow signal and improves the sensitivity of the retrieval, especially for large droplets. However, the specMACS measurements have the highest angular resolution. From a technical perspective it is interesting that a high angular resolution is required to retrieve such large $r_{\text{eff}}$ as retrieved in case study 1, as the cloudbow signal becomes narrower for large $r_{\text{eff}}$ (see Fig. 5 a). To determine the required
angular resolution, Miller et al. (2018) used the Nyquist frequency, which defines the minimum sampling resolution needed to resolve features of an oscillating signal. In addition to the $r_{\text{eff}}$, the required angular resolution depends on the wavelength $\lambda$, with shorter wavelengths requiring a higher angular resolution (shown, e.g., in Fig. 1 a in McBride et al. (2020)). The Nyquist resolution for $\lambda = 670\,\text{nm}$ and $r_{\text{eff}} = 40\,\mu\text{m}$ is approximately 1.5° according to Miller et al. (2018). This is a challenge for some polarimetric instruments because they do not measure with the necessary angular resolution (e.g. POLDER: 4° to 10° (Shang
et al., 2015), AirHARP: 2° (McBride et al., 2020)). RSP measures at an angular resolution of 0.8° and does retrieve $r_{\text{eff}}$ larger than 30 µm, but so far, an example of such large $r_{\text{eff}}$ has not yet been discussed in any study as mentioned in Sinclair et al. (2021). The need of a high angular resolution is no limitation for the specMACS instrument (measurements are binned onto a grid with stepsize 0.3°).

As mentioned above, the large FOV of the specMACS instrument favours the observation of the cloudbow. We assessed how
often the cloudbow retrieval can actually be applied during typical observation conditions. Only measurements that cover the whole scattering angle range 135° to 165° are evaluated (see evaluable stripes in Fig. 7 and Fig. 12). Although we need this special observation geometry, the acquired cloudbow dataset is very large. During daytime with optimal cloudbow conditions (high solar zenith angle), approximately 45 % of the field-of-view of one camera can be analyzed which corresponds to an 8 km wide swath at 10 km flight altitude. We computed the area of the evaluable stripe averaged over all EUREC[4]A measurements

(130 flight hours) including night flights. The analysis showed that when combining the measurements of the two cameras, on average a stripe consisting of 25 % of all pixels of an image can be evaluated. It should be noted, that several EUREC$^4$A HALO flights were partly conducted during nighttime (Konow et al., 2021), during which the observation of the cloudbow is of course impossible. A recent study by Thompson et al. (2022) evaluated the sampled scattering angle range of multi-angle satellite instruments depending on different factors such as solar and view geometry, or location, season, and swath width in

more detail.

A statistical evaluation of all EUREC$^4$A flights, with special attention to the differences in cloud microphysics for the different mesoscale patterns of trade-wind clouds (Sugar, Gravel, Fish, Flower) is planned. Further processing steps will include comparisons of the polarimetric retrieval with bi-spectral retrievals both from specMACS, and from satellites such as MODIS and GOES, and validation of the polarimetric retrieval with in situ measurements. During EUREC$^4$A the British research

airplane Twin Otter and the French airplane ATR-42 collected measurements inside the clouds, which will be compared to our results. Further validation is planned with in situ data of the recent CIRRUS-HL (2021) and HALO-(AC)[3] (2022) campaigns. Moreover, we plan to apply the RFT technique (Alexandrov et al., 2012b) to specMACS measurements in the future, which will be particularly interesting for case studies involving bimodal DSDs. In addition, the retrieval will also be applied to the spatially limited, but very precise measurements of the backscatter glory.

specMACS offers great potential for further evaluations of clouds. In future, the measurements of the different specMACS cameras (polarimetric and hyperspectral) will be combined to retrieve information about the variation of cloud microphysical properties with height inside the cloud, and to identify the thermodynamic phase of the observed clouds. Furthermore, the HALO remote sensing payload makes it possible to deepen the understanding of clouds by combining the measurements of the different instruments.

*Data availability.* The specMACS data used in this study are available upon request from the corresponding author.

*Author contributions.* TK, TZ, BM, LF and VP actively participated in the EUREC$^4$A field campaign. TK, TZ, BM prepared the field campaign, and provided valuable input during the development of the method. TK designed the data file format, developed the stereo method, and provided software for initial use. LV improved the stereo method, and applied it to the data of the EUREC$^4$A field campaign. AW carried out the geometric calibration of the cameras and provided the software code to process and calibrate the raw data. CE provided valuable input
about polarization. BM and CE helped carry out the phase matrix simulations. VP developed the polarimetric cloudbow algorithm, processed and visualized the data and wrote the manuscript with input from all coauthors.

*Competing interests.* Some authors are members of the editorial board of Atmospheric Measurement Techniques. The peer-review process was guided by an independent editor, and the authors have also no other competing interests to declare.

*Acknowledgements.* Thank you, Bjorn Stevens, MPI for Meteorology Hamburg and Markus Rapp, Institut für Physik der Atmosphäre, DLR Oberpfaffenhofen for funding the specMACS contribution to EUREC$^4$A. We thank the EUREC$^4$A project team for collaboration and support and special thanks go to the DLR flight operations team for the planning and the execution of the HALO flights. Furthermore, we thank Andreas Giez, Vladyslav Nenakhov, Martin Zöger and Christian Mallaun for providing the BAHAMAS dataset and valuable information about it. Thank you, Silke Groß and Martin Wirth for providing the WALES lidar data. Many thanks to Alexander Scheiderer and Zhoutong Ma, for developing the cloud mask algorithm. Thank you, Heike Konow and Lutz Hirsch for sharing the HAMP radar data and for providing valuable information about the data. The data used in this publication was gathered in the EUREC4A field campaign and is made available through Meteorologisches Institut, Ludwig-Maximilians-Universität München. EUREC$^4$A is funded with support of the European Research Council (ERC), the Max Planck Society (MPG), the German Research Foundation (DFG), the German Meteorological Weather Service (DWD) and the German Aerospace Center (DLR).

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
