# Peer review of "High spatial resolution retrieval of cloud droplet size distribution from polarized observations of the cloudbow"

_Atmospheric Measurement Techniques, 2022_

## Referee Comment (RC2)

This work demonstrates a legacy polarimetric cloud retrieval algorithm on data from a new dual-imaging polarimeter instrument, specMACS. specMACS is a wide-field, division of focal plane polarimeter. 3-band VIS wavelength selection is done using a Bayer-filter-like scheme at the detector. The polarimeter has two identical cameras looking slightly off-nadir such that their FOVs have a significant overlap. This design produces an effective FOV larger than both cameras alone.

specMACS co-registered, multi-angle data over large stratiform clouds and trade wind cumulus were used to explore cloud information content at scales ~100m. These retrievals were done over a wide spatial field, in the overlap region between both cameras. This work is a spiritual successor to McBride et al. (2020), which was the first to do a similar study using cloud measurements from the AirHARP polarimeter. This work improves on that study in several ways: (1) a stereo cloud height is determined for each cloud pixel, and (2) the authors claim retrieval sensitivity in effective radius ($r_{eff}$) up to 40um and at a spatial resolution two times smaller than results in the McBride paper, and (3) an application to small cumulus clouds is shown. These factors and others may provide new opportunities to tease out cloud processes with specMACS data.

As polarimetric instrument development continues, papers like these help track the state of the field. They will be used to inform new instrument designs, missions, and algorithms. Therefore, I recommend this exciting paper for publication in AMT. However, I suggest a mix of specific and minor revisions.

Specific Comments

(1) The opening paragraph needs more discussion on the current state of cloud and climate science: their non-uniform global distribution, both vertically and horizontally, thermodynamic phase differences (ice vs. water), and current challenges in comparing remote sensing/in-situ measurements/retrievals, cloud simulations, and global climate models. These are just examples, but please add a few extra sentences that put the paper in stronger context with the field.

The connection between aerosols and clouds and the potential benefit of using polarmetric measurements in aerosol-cloud studies is missing in the Introduction. This is a hot field of current cloud research and polarimeters like specMACS are highly relevant to this topic. Please add a few sentences about this.

(2) One of the major results of this paper is the ability to retrieve $r_{eff}$ ~40um from specMACS cloud data. This capability is highly attractive for future climate applications and missions. This can be a challenge for some polarimeters and retrievals done on their data. For this kind of retrieval, some polarimetric instruments are capped in the upper bound of retrieved $r_{eff}$ due to limitations in view zenith angle density (Miller et al. 2018, citation below). specMACS may get around this limitation with its dual-camera design and access to a second set of retrievable pixels and geometry for the same cloud target. This is important to mention. Also, more details about how the specMACS design and sampling directly compares to other, similar cloud measuring instruments (specifically AirHARP and RSP) would be valuable.

Miller, D. J., Zhang, Z., Platnick, S., Ackerman, A. S., Werner, F., Cornet, C., and Knobelspiesse, K.: Comparisons of bispectral and polarimetric retrievals of marine boundary layer cloud microphysics: case studies using a LES–satellite retrieval simulator, Atmos. Meas. Tech., 11, 3689–3715, https://doi.org/10.5194/amt-11-3689-2018, 2018.

This paper would also benefit from a short, quantitative discussion of retrieval uncertainty (Qual and RMSE). How well can one use specMACS data to reliably retrieve large $r_{eff}$ and $v_{eff}$? An accompanying sub-figure similar to Figure 7a and b that shows the spatial distribution of the best-fit RMSE may support this discussion.

(3) In many areas, this paper has colons (:) when a period would be more effective. Please look through the document and make revisions as needed.

Technical Corrections

**Abstract**

Please put the conclusion of this paper at the end of the abstract.

**Section 1**

Line 21: What is "extreme precipitation"?
Line 22: "future temperature changes" is vague. The IPCC AR6 is specific in defining how future climates may be influenced by changes in sea surface temperature, global mean air surface temperature, or other similar measures. Please be specific.

Line 39: Large phrases like "droplet size distribution" can be reduced to conventional acronyms, like DSD. Also, since effective radius and effective variance are defined as $r_{eff}$ and $v_{eff}$ in Lines 43/44, please can use these abbreviations going forward in the paper. Other phrases can be simplified too - like "degree of linear polarization" to DOLP.

Lines 42 and 97: Remove the "e.g."

Line 51: "it has some difficulties which are mainly related to 3-D effects occurring especially in inhomogeneous cumulus cloud fields" should change to "has known biases in the presence of 3-D effects and spatial inhomogeneity." Spatial inhomogeneity impacts the retrieval at some level for all cloud types, and those biases aren't always related to 3-D effects.

Line 53: "Furthermore, retrieving the effective variance of the cloud droplet size distribution is not possible." Please add a citation and/or elaborate.

Line 58, 134, 239, 292, 345: please remove "so-called" in all instances. In most cases, these techniques actually go by these names (i.e. scattering matrix is the name of that matrix, wire-grid polarizer is the official term for that kind of polarizer). In Line 59, it would be a stronger sentence as "Based on polarized observations of the cloudbow, a new kind of DSD retrieval was developed."

Line 73: The following two sentences are somewhat redundant. Something like "Furthermore, the $v_{eff}$ of the cloud DSD is derived in the polarimetric retrieval. This parameter may be directly linked to entrainment and mixing processes at the cloud top." would be stronger.

Line 91: The ~2um $r_{eff}$ bias between the MODIS bi-spectral and polarimetric DSD techniques was found in other earlier studies as well, largely due to information content differences in location of the cloud DSD creating the signal, retrieval resolution, inhomogeneity in the pixel, and choice of SWIR band used in the bi-spectral retrieval. Please cite as necessary:

Alexandrov et al. (2015, 2016)
Breon and Doutriaux-Boucher (2005)
and

Di Noia, A., Hasekamp, O. P., van Diedenhoven, B., and Zhang, Z.: Retrieval of liquid water cloud properties from POLDER-3 measurements using a neural network ensemble approach, Atmos. Meas. Tech., 12, 1697–1716, https://doi.org/10.5194/amt-12-1697-2019, 2019.

Line 97: Please re-word this sentence for clarity, something like "RSP data can provide a bi-spectral and a polarimetric $r_{eff}$ from the same cloud target, due to spectral coverage from VIS to SWIR and along-track, co-located multi-angle sampling."

**Section 2**

Line 133 (and Figure 1). I appreciate the relative spectral response figure added to the manuscript. It would be helpful to also include the center wavelength and bandwidth (FWHM) of the three filters in the text itself. The typical convection is to write it as wavelength (bandwidth), like 440 (15) nm.

Line 142: Please reword this statement - the degree of linear polarization (DOLP) describes the fraction of the incoming light that is linearly polarized. Q and U also quantitatively describe the linear polarization.

Line 152: Remove the comma between "advantage, that"

Line 156: "For further analysis, each measured Stokes vector is rotated into the scattering plane (Hansen and Travis, 1974) and we only evaluate Q." Although this work focuses on solar principal plane (SPP) geometries (typically a narrow line of pixels in an observation), there will likely be non-zero U values for cloud targets located off the SPP in the spatial field retrieval. Were there cloud targets with non-negligible U values? If so, how does that contribute to the interpretation of the retrieval results?

Figure 2 caption: Please reword for clarity - "The primary bow of the cloudbow is visible in the degree of linear polarization as a bright ring at a scattering angle of about 140°"

**Section 3**

Line 173: "This method can easily be applied to any cloudbow observations, including those from commercial cameras, but it requires averaging over a large area." Please describe why is this less desirable than the co-located, along-track method presented in this paper.

Line 175: Please change "we fly over it" to "specMACS images the scene".

Line 181: "In a final step, a look-up table (LUT) of cloudbow signals for different cloud droplet size distributions.." Please be specific that a simulated Mie LUT is used for this comparison.

Figure 3: The alpha angle in the figure is missing a zenith line coming directly from the plane, please add this in.

Figure 4: The colorbar is very large and doesn't apply to (d), so it would be cleaner visually if it was condensed and placed under (c). The caption would need to reflect this change as well. Please adapt Figure 10 similarly.

Line 240: Please remove the (single scattering). This is redundant with line 239.

Line 241: The polarized phase function is directly proportional to the measured polarized radiance Q, under the assumption of single scattering. $P_{12}$ is not an approximation of Q. Please revise.

Line 245: Effective variance determines the amplitude of the secondary maxima/minima (see Figure 5b), which is where the information content lies (not number of peaks). Please revise this sentence.

Line 254: Please add the word unimodal or monomodal before "gamma distribution".

Line 280 and 287: Thanks for changing Eq. (6) and (8) to the typical/original convection!

Line 284: It has also been shown in Alexandrov et al. (2012a) that increasing cloud and/or aerosol optical thickness can impart a linear slope on Q. Reidi et al. (2010) shows that cirrus ice polarization signal is approximately linear in the rainbow region, too. These contaminations can also be accounted for by B and C terms in Eq. (6).

Riedi, J., Marchant, B., Platnick, S., Baum, B. A., Thieuleux, F., Oudard, C., Parol, F., Nicolas, J.-M., and Dubuisson, P.: Cloud thermodynamic phase inferred from merged POLDER and MODIS data, Atmos. Chem. Phys., 10, 11851–11865, https://doi.org/10.5194/acp-10-11851-2010, 2010.

Figure 6: The presentation of the "number of measurements" is a little confusing. In the plots, it almost looks like an uncertainty but it's a bit more complex than that and it blurs the overall message in my view. When the slope of the Q_fit signal is large in a small scattering angle range (see 6a between 142-147 degrees), it is hard to differentiate the box values from the measurement/fit lines.

I recommend to replace the "number of measurements" pixels in the plots with errorbars that correspond to the angular measurement. This goes for Figures 9 and 15 as well.

Section 4

Line 376: Please remove the comma between "example, to"

Line 378: Please remove the e.g. and commas.

Line 380: Please change to "In the case of small cumulus clouds, a precise geolocalization is important for image-to-image tracking."

Line 393: Please define 100m as a "target unit" here. This is definition referenced later in Line 401, but in an indirect way and was confusing on a blind read.

Figure 10: See comment for Figure 4.

Figure 10c: Why does the interpolation at the bottom right look artificial? Since it is not discussed in the text, I recommend to screen out this data and leave it unassessed (i.e. as it looks in 10a).

Line 423: If the error due to incorrect geolocalization is yet be estimated quantitatively, how is it that the cloudbow retrieval is "affected [by it] to a much lesser degree"? This is confusing. Please elaborate.

Line 433: It is clear in 12b that the majority of 0.3+ veff values are "tracing" the lower boundary of the overlap between polA and polB cameras. 12f shows a big spike at the 0.32 $v_{eff}$ bin, which is the very edge of the Mie LUT described on Line 269. This together suggests that these retrievals are artificially converging (either due to noise or errors in geolocalization) and may not be valid, even if they pass the Qual and RMSE. I suggest removing this $v_{eff}$ bin from the Figure 7f and 12f plots (and corresponding data points from the 7b and 12b maps) to focus the discussion on the valid larger $v_{eff}$ that do exist in the data.

Line 458: McBride et al. (2020) was also careful to note that the subpixel $r_{eff}$ and $v_{eff}$ distribution that contributed to their larger-scale wide DSD result could also be impacted by cloud height geo-registration (they used a granule-wide average, not pixel-by-pixel as in this study). I suggest to change the "does" to "may" in this sentence.

**Section 5**

Line 468: Remove the comma between "cloud, that" and change "are" to "may be". This has not been shown in this study, only suggested.

Line 470: Add a comma between "layer very"

---

## Author Comment (AC1)

**Response to review comment 1**

**Manuscript:** AMT-2022-245
**Title:** High spatial resolution retrieval of cloud droplet size distribution from polarized observations of the cloudbow
**Authors:** Veronika Pörtge, Tobias Kölling, Anna Weber, Lea Volkmer, Claudia Emde, Tobias Zinner, Linda Forster and Bernhard Mayer

**Response to report #1**

We thank Bastiaan van Diedenhoven for his comments and suggestions which we address in the following. The authors' answers are printed in italics, and with gray background.

**Summary:**

This is a review of the paper titled "High spatial resolution retrieval of cloud droplet size distribution from polarized observations of the cloudbow" submitted to AMT by Pörtge et al. The paper describes the measurements of polarimetric cameras which are part of the specMACS instrument deployed on the HALO aircraft. The data processing is described, as well as the application of polarimetric cloudbow retrievals of the cloud top droplet size distribution. A few case studies are described in detail.

The paper is generally well written. The data and the results are very interesting and show a lot of potential. I recommend publication of the paper after addressing some minor comments and questions listed below and some corrections to the text and figures.

**General comments:**

1. The data is corrected for the displacement of the clouds using ERA5 wind fields, as mentioned in section 3.2. As this correction seems not entirely trivial to me, I would suggest adding a bit more detail. For example, the ERA5 resolution is much coarser than the observations, so are the ERA5 fields interpolated to the observation locations? If not, I would expect strange effects at the ERA5 boundaries. (Please give the horizontal and vertical resolution of ERA5.) Also, how is the vertical variation of the wind field taken into account, as the cloud top height is not known yet at this step?

   > → *Thanks for pointing this out. This is indeed an important information and we added the following text to the section:*
   >
   > *"To correct for horizontal displacements of the cloud, the method was extended to include data of the horizontal wind from the ERA5 reanalysis dataset (Hersbach et al., 2020, 2018). The dataset has an hourly temporal resolution, a horizontal resolution of 0.25° × 0.25°, and 37 vertical levels from the surface to 1 hPa. During EUREC4A, clouds were typically observed at a vertical altitude of 1 km to 2 km where the ERA5 dataset has a vertical grid spacing of about 250 m. For testing purposes, we also used the ERA5 data on the original model levels with 137 vertical levels, and a higher vertical resolution (about 150 m at a vertical altitude of 1 km to 2*

*km) but this had no significant effect on the derived stereo heights. First, the stereo method is performed without additional wind information, and the 3-D coordinates of the identified pixels (stereo points) are retrieved. Then, the ERA5 data are interpolated to these coordinates to extract the corresponding wind data. The stereo method is performed again, but this time taking the wind data into account. The whole process is iteratively repeated 5 times, each time updating the wind data with the ERA5 wind interpolated to the heights and locations of the previously found stereo points. Further increasing the number of iterations did not notably change the results."*

2. If I understand correctly, the cloud top heights are first determined for cloud (parts) that show prominent features and those results are then interpolated to the rest of the field. Again, this interpolation seems not entirely trivial to me and more details are needed. Is it just a simple 2D interpolation in lat/lon space to the nearest points?

*→ The interpolation is now explained in more detail:*

*"The cloud top heights from the single points of the stereographic method are interpolated to the entire image (Fig. 4 c). The interpolation process first consists of a linear interpolation of the stereo pixels onto all image pixels inside the convex hull of the stereo pixels. Then, the regions outside the convex hull of the original stereo points are filled by a nearest neighbor interpolation. The resulting cloud top heights are assigned to the selected cloud targets."*

3. In section 4.1., the detection of large cloud drops are discussed. In line 325 it is stated that the detection of these large drops "is confirmed by high reflectivity values of the polarimetric Ka-band MIRA-35 cloud radar measurements". However, this radar is sensitive to rain drops that are much larger than the large cloud drops observed by specMACS. These cases likely have bi-modal size distributions, as discussed in the manuscript, but the large drops mode is not a rain mode, as suggested in line 333, but more likely drops in the 'size gap' as discussed in line 345. I think the discussion in the manuscript is a bit confusing now, somewhat suggesting the large drops are rain drops. I suggest rewriting the discussion to emphasize the distinction between rain and the large drops. Sinclair et al. (2021) also discussed the association between large cloud drops at cloud top observed with a polarimeter and rain detected by radar.

*→ We addressed your comments by changing the discussion to:*

*"The retrieved reff values of the higher cloud are very large. To better understand the cloud field and the large reff values, we looked at radar measurements of the polarimetric Ka-band MIRA-35 cloud radar of the HAMP instrument onboard HALO (Mech et al., 2014; Konow et al., 2021). The radar measurements from 16:47:00 to 16:48:30 UTC are shown in Fig. 8 along with a push-broom like image of the specMACS measurements and an indication of the HAMP radar field-of-view within the specMACS image. Within the high cloud from 16:47:00 to 16:48:15 the radar shows bands of enhanced reflectivity > 0 dBz and positive fall speeds (not shown). This likely corresponds to sedimenting droplets. Together with our observation of droplet sizes clearly larger than the usual cloud droplet size range (<15 µm) this points to drizzle development and we may see impacts of precipitation formation deeper in the cloud within the polarimetric signal originating from cloud top. Although our technique is not able to observe the precipitation droplet range (>100 µm) directly it is still sensitive to the*

*intermediate size range below a possible drizzle droplet mode. This case study is particularly interesting, as the retrieved reff lie within the size gap where neither the diffusional growth, nor growth by collision-coalescence is effective (Grabowski and Wang, 2013). The recent study by Sinclair et al. (2021) discussed the correlation between large cloud droplets detected by the RSP polarimeter and rain observed with a radar in great detail and found that the estimated cloud top precipitation rates are strongly correlated to radar derived precipitation rates and rainwater paths. For our polarimetric technique, it is necessary to make an assumption on the shape of the DSD. Currently, we use a monomodal gamma distribution for this purpose. In Alexandrov et al. (2012b) it was shown that for clouds with a bimodal DSD (e.g., due to drizzle), the polarimetric retrieval based on monomodal DSDs is biased towards the dominant mode. To overcome this problem, the rainbow fourier transformation (RFT) was developed, that retrieves the DSD without any assumptions on the number of modes of the distribution (Alexandrov et al., 2012b). When comparing the polarimetric technique to the traditional bi-spectral retrieval, it should be noted, that bi-spectral retrievals are (normally) also based on simulations with monomodal DSDs (Platnick et al., 2017) which tends to underestimate the true reff and has been investigated in several studies (e.g. Zinner et al., 2010; Zhang et al., 2012; Zhang, 2013)."*

4.  In figure 13 an effective radius "profile" is shown. It is probably good to point out the profile is not a profile of effective radius inside a cloud, but statistics of effective radius at different cloud top heights. Some more details of how this profile is obtained would be good. For the interpretation, you may want to refer to Rosenfeld and Lensky (1998).
    https://journals.ametsoc.org/view/journals/bams/79/11/1520-0477_1998_079_2457_sbiipf_2_0_co_2.xml

*→Thanks for pointing that out. We added an explanation of how we derive the vertical "profile" and referred to Rosenfeld and Lensky (1998) for the interpretation:*

*"This dependence of the reff on the cloud top height is presented in more detail in Figure 13. Here, the derived reff of all individual clouds of the case study are plotted against the corresponding cloud top heights (as in Rosenfeld and Lensky (1998)). We refer to this plot as a vertical profile, although it does not show the actual reff profile within a single cloud. The idea is that the individual clouds of the cloud field are captured at different stages of their vertical growth. It is then assumed that the retrieved reff which is sampled at the cloud top is representative of the actual reff at the same height inside a single cloud. This assumption applies only to non-precipitating clouds. By combining the measurements of the individual clouds at different stages of their vertical development, a vertical profile is constructed, which is assumed to correspond to the vertical profile of a single growing cloud. The figure shows a strong increase of reff from about 5 µm at a height of 800 m to 9 µm at 1350 m. This rapid growth of droplets with height may indicate the dominance of growth through coalescence and is typical for maritime clouds. Rosenfeld and Lensky (1998) refer to this zone as the "droplet coalescence growth zone".*

5.  In line 431-439 some reasons for the detection of large effective variance values, mainly at the cloud edges. It is suggested that errors in the aggregation of angles may have caused the

cloudbow signal to be distorted. But would these cases not be filtered out by the RMSE and Qual requirements?

*→ We again checked the targets with large veff. We think that there are some targets that show such large veff values which is why we do not want to rigorously filter out all veff > 0.32 as suggested by Reviewer #2.*

a) *Some signals actually came from ocean targets. In this case, the structure of the aggregated signal is relatively linear and the fit function can be fitted perfectly to the signal by keeping the parameter A (which scales the phase function) in the fitting function small. The fit has a very small RMSE, and as Qual ~ 1/RMSE this might also lead to a relatively high Qual index (at least > 2 as our original Qual filter threshold) and the signal is not filtered out.*

b) *Other signals (especially at cloud edges) did probably suffer from errors in the aggregation of angles. These did show a cloudbow signature, but the signal was not very clear, which could result from a mixture with ocean measurements. We increased the Qual filter threshold from 2 to 4. This did also filter out some targets, that are located quite central within the clouds, but where the cloud showed a lot of variability (shadows) and where an error in the aggregation of angles also could have a large effect.*

c) *There were some targets with veff = 0.32 which were located quite central within clouds (especially in case study 1) where the cloud did not show a lot of variability. The signals looked good and we think that these targets do have a veff of 0.32 or even larger. Such targets were not filtered out by the new Qual threshold of 4 and we decided to keep them in our results.*

**Minor corrections:**

- Line 24: These two sentences basically say the same thing twice. I suggest to remove the first sentence and move the IPCC reference to the second.

*→ Changed as suggested*

- Line 53: Add "using the bi-spectral technique " after "is not possible".

*→ Changed to:*

*"Furthermore, the bi-spectral technique does not provide information about veff (Nakajima and King, 1990)."*

- Line 72: this should be "singly-scattered photons".

*→ Changed as suggested*

- Line 75: I suggest changing the order of words into "which is a parameter that is directly linked to entrainment and mixing processes."

*→ Changed to:*

*"Furthermore, the veff of the DSD is derived in the polarimetric retrieval. This parameter may be directly linked to entrainment and mixing processes at the cloud top."*

- Line 80: An example of cloudbow retrieval applied to airMSPI is given by Xu et al. https://agupubs.onlinelibrary.wiley.com/doi/10.1002/2017JD027926

→ *We added Xu et al. 2018 as a reference.*

- Line 99: Please place a hyphen between 'bias' and 'adjusted'

→ *Changed as suggested*

- Line 205: I think "from" should be "of" in this sentence.

→ *Changed as suggested*

- Line 235: "the radiance measurement is binned". Do you mean Stokes parameter Q?

→ *Yes, we mean the Stokes parameter Q as in the previous sentence. To make it clear that the two sentences belong together, we have changed them as follows:*

*"The individual measurements of the same target of the Stokes parameter Q are combined to generate the aggregated polarized radiance measurement. For further processing, the aggregated measurement is binned onto a scattering angle grid with a step size of 0.3°."*

- Line 240: "The P12 element is also called the polarized (single scattering) phase function and is a good approximation for the measured polarized radiance Q". The absolute value of P12 itself is not a good approximation of measured Q, but the relative variation is. Also, it is not Q, but Q rotated to the scattering plane. Please rewrite this sentence accordingly.

→ *We changed the sentence to:*

*"The P12 element is also called the polarized phase function and is approximately proportional to the measured polarized radiance Q in the scattering plane (Bréon and Goloub, 1998)."*

- Line 245: The effective variance also determines the width of the secondary minima.

→ *Thank you for pointing this out, we added "and widths" to the sentence:*

*"The veff, however, determines the amplitude and widths of secondary minima of the radiance distribution but has only a small effect on the position of the minima."*

- Line 294: Please place Eq. 8 here in the sentence.

→ *We added the reference to Eq. 8 to the first sentence and removed it from the second:*

*"As a second quality measure, we calculate the quality index "Qual" as in Equation 8 (first defined by Bréon and Doutriaux-Boucher (2005)). This is the ratio between the variability of the measurement, which corresponds to the squared amplitude of the cloudbow (A·P 12 ), and the RMSE of the fit."*

- Line 342: Please remove the comma after 'statement'

> *→We restructured the whole paragraph (see specific comment 3).*

- Figure 7 and 10: Please add some information about the (approximate?) scale to the figures. Either a scale bar or x and y axis labels.

> *→Changed as suggested and added explanatory text, on why the image is distorted.*
>
> *"Fig. 7 g) shows the RGB image of the measurement from the polB camera. The labels on the four sides of the image indicate the distances between the neighboring corners of the image. It is noticeable that the side lengths of the top (14.44 km) and bottom (27.02 km) differ greatly. This happens, because the camera is installed at a slight angle in the across-track direction and therefore, the lower part of the image covers a much larger distance in the along-track direction. This is also the case for the measurements of the polA camera, but here, the upper part of the image covers a larger distance."*

- Line 378: I suggest adding Zhang et al. 2012 as an additional reference. https://agupubs.onlinelibrary.wiley.com/doi/10.1029/2012JD017655

> *→We added Zhang et al. 2012 as a reference.*

- Line 453: I suggest adding "to interpret" after "more difficult".

> *→Changed as suggested*

- Line 475: A period is missing after "0.07".

> *→Changed as suggested*

---

## Author Comment (AC2)

**Response to review comment 2**

**Manuscript:** AMT-2022-245
**Title:** High spatial resolution retrieval of cloud droplet size distribution from polarized observations of the cloudbow
**Authors:** Veronika Pörtge, Tobias Kölling, Anna Weber, Lea Volkmer, Claudia Emde, Tobias Zinner, Linda Forster and Bernhard Mayer

**Response to report #2**

We thank referee #2 for his/her comments and suggestions which we address in the following. The authors' answers are printed in italics, and with gray background.

**General comments:**

This work demonstrates a legacy polarimetric cloud retrieval algorithm on data from a new dual-imaging polarimeter instrument, specMACS. specMACS is a wide-field, division of focal plane polarimeter. 3-band VIS wavelength selection is done using a Bayer-filter-like scheme at the detector. The polarimeter has two identical cameras looking slightly off-nadir such that their FOVs have a significant overlap. This design produces an effective FOV larger than both cameras alone.

specMACS co-registered, multi-angle data over large stratiform clouds and trade wind cumulus were used to explore cloud information content at scales ~100m. These retrievals were done over a wide spatial field, in the overlap region between both cameras. This work is a spiritual successor to McBride et al. (2020), which was the first to do a similar study using cloud measurements from the AirHARP polarimeter. This work improves on that study in several ways: (1) a stereo cloud height is determined for each cloud pixel, and (2) the authors claim retrieval sensitivity in effective radius (reff) up to 40um and at a spatial resolution two times smaller than results in the McBride paper, and (3) an application to small cumulus clouds is shown. These factors and others may provide new opportunities to tease out cloud processes with specMACS data.

As polarimetric instrument development continues, papers like these help track the state of the field. They will be used to inform new instrument designs, missions, and algorithms. Therefore, I recommend this exciting paper for publication in AMT. However, I suggest a mix of specific and minor revisions.

**Specific comments:**

1.

   a) The opening paragraph needs more discussion on the current state of cloud and climate science: their non-uniform global distribution, both vertically and horizontally, thermodynamic phase differences (ice vs. water), and current challenges in comparing remote sensing/in-situ measurements/retrievals, cloud simulations, and global climate models. These are just examples, but please add a few extra sentences that put the paper in stronger context with the field.

> → *We added the following text to the introduction:*
> *"Clouds are complex phenomena, and understanding them is a challenging research*

*topic. They can form in almost any region of the Earth and appear at different heights in the atmosphere. What makes them so complicated is, e.g., their high variability both in space and time. In addition, cloud particles have complex microphysical properties and exist in different thermodynamic phases (liquid, ice, supercooled liquid). This significantly impacts the radiative properties of a cloud. The study of clouds becomes even more difficult since aerosols must also be considered to better understand clouds. Aerosols serve as cloud condensation nuclei and affect clouds directly by changing the cloud droplet number concentration, but also by, e.g., suppressing rain, which in turn can change the cloud lifetime (Albrecht, 1989). Simulating clouds in models is challenging, not only because of the issues mentioned above but also because clouds occur at different scales. Their size can be as small as a few meters or as large as hundreds of kilometers, which is an issue for models as they are always limited by their resolution. Although there are cloud-resolving models, that substantially help in understanding clouds, such models are computationally very expensive and still rely on parameterizations which are subject to uncertainties (Satoh et al., 2019). At the same time, measuring clouds is equally difficult. In situ measurements accurately represent the atmospheric state of a few cubic centimeters, but this may not be representative for the cloud as a whole. Observing clouds by remote sensing instruments suffers from retrieval uncertainties and in general, improving models based on observations is not a straightforward task. Although the understanding of clouds has improved due to more and better observations as well as new cloud modeling approaches, the influence of clouds remains a large uncertainty in predicting future climate (Forster et al., 2021). This is why there is a great interest in extending our knowledge of clouds."*

b) The connection between aerosols and clouds and the potential benefit of using polarmetric measurements in aerosol-cloud studies is missing in the Introduction. This is a hot field of current cloud research and polarimeters like specMACS are highly relevant to this topic. Please add a few sentences about this.

*→ We added the following text to the introduction which highlights the benefit of using polarimetric measurements for aerosol studies and lists some of the planned satellite missions:*

*"The additional information from polarimetric measurements is also advantageous when it comes to studying aerosols (Remer et al., 2019). Aerosols and clouds have different angular polarimetric signatures (e.g., Emde et al., 2010), which can be exploited to distinguish between aerosols and clouds. Furthermore, theoretical studies showed that aerosol properties can be retrieved from polarimetric measurements with sufficient accuracy for climate research (e.g., Mishchenko and Travis (1997); Hasekamp and Landgraf (2007)). For instance, the simultaneous characterization of cloud properties and properties of aerosol above clouds (Knobelspiesse et al., 2011), or of aerosol between clouds (Hasekamp, 2010) is possible when using multi-angle polarimetric measurements. Obtaining polarization data from space is therefore desirable to improve the global picture of the atmosphere concerning both cloud and aerosol properties, and to quantify aerosol-cloud interactions. For this reason, several satellite missions with polarimetric instruments onboard will soon be launched or are already in space. The PACE (Plankton, Aerosol, Cloud, ocean Ecosystem) mission will*

2.

*be a polar-orbiting satellite that will deploy two polarimeters for cloud and ocean retrievals (Remer et al., 2019), the 3MI instrument (Multi-view Multi-spectral Multi-polarization Imager) will be part of the payload of the MetOp-SG satellite (Fougnie et al., 2018), and the MAIA instrument (Multi-Angle Imager for Aerosols) (Diner et al., 2018) will help to characterize particulate matter in air pollution, to name a few of the planned satellite instruments. The various existing polarimetric instruments, and those under development are listed in Dubovik et al. (2019). This shows that the development of polarimetric instruments is an active research focus and polarimetric airborne instruments are highly useful in investigating appropriate instrument design, satellite mission planning, or retrieval techniques."*

a) One of the major results of this paper is the ability to retrieve reff ~40um from specMACS cloud data. This capability is highly attractive for future climate applications and missions. This can be a challenge for some polarimeters and retrievals done on their data. For this kind of retrieval, some polarimetric instruments are capped in the upper bound of retrieved reff due to limitations in view zenith angle density (Miller et al. 2018, citation below). specMACS may get around this limitation with its dual-camera design and access to a second set of retrievable pixels and geometry for the same cloud target. This is important to mention.

> →*Thanks for pointing out that the high angular resolution is another feature of the instrument. We investigated our case study 1 (Fig. 7) with regard to the angular resolution by reducing the resolution from 0.3° to 1.5°. A resolution of 1.5° is approximately the required angular resolution to successfully resolve the cloudbow signal of cloud droplets with reff = 40 µm at λ = 0.67 µm according to Miller et al. (2018). As expected, the spatial distribution of the DSD of the lower cloud (with reff < 15 µm) did not change, but for the upper cloud (reff = 15 − 40 µm) the retrieved results changed when using the lower angular resolution. The derived variance increased (90ᵗʰ percentile of the distribution increased from 0.2 to 0.24) as well as the effective radius (90ᵗʰ percentile increased from 30.99 µm to 31.57 µm with the highest change occuring for cloud targets with reff > 25 µm). All mentioned values come from the retrieval results of the green channel.*
>
> *See the next comment (comment 2b) for the corresponding changes in the text.*

b) Also, more details about how the specMACS design and sampling directly compares to other, similar cloud measuring instruments (specifically AirHARP and RSP) would be valuable. Miller, D. J., Zhang, Z., Platnick, S., Ackerman, A. S., Werner, F., Cornet, C., and Knobelspiesse, K.: Comparisons of bispectral and polarimetric retrievals of marine boundary layer cloud microphysics: case studies using a LES–satellite retrieval simulator, Atmos. Meas. Tech., 11, 3689–3715, https://doi.org/10.5194/amt-11-3689-2018, 2018.

> →*We added a table which summarizes the main differences between specMACS, AirHARP and RSP and an explanatory text to the "Discussion and Conclusion" chapter:*

[revised manuscript text omitted]

c)  This paper would also benefit from a short, quantitative discussion of retrieval uncertainty (Qual and RMSE). How well can one use specMACS data to reliably retrieve large reff and veff? An accompanying sub-figure similar to Figure 7a and b that shows the spatial distribution of the best-fit RMSE may support this discussion.

> → *We added the RMSE spatial distribution and frequency distribution to the Figures 7, 12, 14, and added the RMSE and Qual values to the Figures (6, 9, 15) that show the aggregated Q-measurements together with the fit.*

3.  In many areas, this paper has colons (:) when a period would be more effective. Please look through the document and make revisions as needed.

> → *Changed as suggested.*

**Technical Corrections:**

**Abstract**

- Please put the conclusion of this paper at the end of the abstract.

> → *Changed as suggested.*

**Section 1**

- Line 21: What is "extreme precipitation"

> → *We added the definition of extreme precipitation according to the IPCC glossary:*
>
> *"Secondly, clouds can produce precipitation that strongly affects our lives, especially in the case of extreme precipitation, which is characterized by its very high magnitude and its very rare occurrence at a specific location (IPCC, 2021)."*

- Line 22: "future temperature changes" is vague. The IPCC AR6 is specific in defining how future climates may be influenced by changes in sea surface temperature, global mean air surface temperature, or other similar measures. Please be specific.

  *→We changed the sentence to be more general using the phrase "future climate", which was our original intention and changed the reference to Forster et al., 2021, which is the specific IPCC chapter, that shows the influence of clouds on climate:*

  *"Although the understanding of clouds has improved due to more and better observations as well as new cloud modeling approaches, the influence of clouds remains a large uncertainty in predicting future climate (Forster et al., 2021)."*

- Line 39: Large phrases like "droplet size distribution" can be reduced to conventional acronyms, like DSD. Also, since effective radius and effective variance are defined as reff and veff in Lines 43/44, please can use these abbreviations going forward in the paper. Other phrases can be simplified too - like "degree of linear polarization" to DOLP.

  *→Changed as suggested.*

- Lines 42 and 97: Remove the "e.g."

  *→ Changed as suggested.*

- Line 51: "it has some difficulties which are mainly related to 3-D effects occurring especially in inhomogeneous cumulus cloud fields" should change to "has known biases in the presence of 3-D effects and spatial inhomogeneity." Spatial inhomogeneity impacts the retrieval at some level for all cloud types, and those biases aren't always related to 3-D effects.

  *→ Changed as suggested.*

- Line 53: "Furthermore, retrieving the effective variance of the cloud droplet size distribution is not possible." Please add a citation and/or elaborate.

  *→ We adjusted the text to:*

  *"Furthermore, the bi-spectral technique does not provide information about v eff (Nakajima and King, 1990)."*

- Line 58, 134, 239, 292, 345: please remove "so-called" in all instances. In most cases, these techniques actually go by these names (i.e. scattering matrix is the name of that matrix, wire-grid polarizer is the official term for that kind of polarizer). In Line 59, it would be a stronger sentence as "Based on polarized observations of the cloudbow, a new kind of DSD retrieval was developed."

  *→ Changed as suggested.*

- Line 73: The following two sentences are somewhat redundant. Something like "Furthermore, the veff of the cloud DSD is derived in the polarimetric retrieval. This parameter may be directly linked to entrainment and mixing processes at the cloud top." would be stronger.

  *→ changed as suggested*

- Line 91: The ~2um reff bias between the MODIS bi-spectral and polarimetric DSD techniques was found in other earlier studies as well, largely due to information content differences in location of the cloud DSD creating the signal, retrieval resolution, inhomogeneity in the pixel, and choice of SWIR band used in the bi-spectral retrieval.

  Please cite as necessary:

  Alexandrov et al. (2015, 2016)

  Breon and Doutriaux-Boucher (2005) and

  Di Noia, A., Hasekamp, O. P., van Diedenhoven, B., and Zhang, Z.: Retrieval of liquid water cloud properties from POLDER-3 measurements using a neural network ensemble approach, Atmos. Meas. Tech., 12, 1697–1716, https://doi.org/10.5194/amt-12-1697-2019, 2019.

  → *We added the mentioned references to the text:*

  *"There are several other studies, such as by Bréon and Doutriaux-Boucher (2005), Di Noia et al. (2019) or Alexandrov et al. (2015) that compared reff obtained from polarized measurements with bi-spectral results and found similar biases. The differences could largely be attributed to the different penetration depths of the SWIR band compared to the polarized signal, to differences in retrieval resolution, and to 3-D radiative transfer effects."*

- Line 97: Please re-word this sentence for clarity, something like "RSP data can provide a bi-spectral and a polarimetric reff from the same cloud target, due to spectral coverage from VIS to SWIR and along-track, co-located multi-angle sampling."

  → *Changed as suggested.*

**Section 2**

- Line 133 (and Figure 1). I appreciate the relative spectral response figure added to the manuscript. It would be helpful to also include the center wavelength and bandwidth (FWHM) of the three filters in the text itself. The typical convection is to write it as wavelength (bandwidth), like 440 (15) nm.

  → *We added the information to the text:*

  *"The spectral channels have center wavelengths (bandwidths) of approximately 620 nm (66 nm), 546 nm (117 nm), 468 nm (82 nm) (determined by a gaussian fit), and the normalized spectral response functions of each color channel are shown in Fig. 1 b)."*

- Line 142: Please reword this statement - the degree of linear polarization (DOLP) describes the fraction of the incoming light that is linearly polarized. Q and U also quantitatively describe the linear polarization.

  → *Changed as suggested.*

- Line 152: Remove the comma between "advantage, that"

  → *Changed as suggested.*

- Line 156: "For further analysis, each measured Stokes vector is rotated into the scattering plane (Hansen and Travis, 1974) and we only evaluate Q." Although this work focuses on

solar principal plane (SPP) geometries (typically a narrow line of pixels in an observation), there will likely be non-zero U values for cloud targets located off the SPP in the spatial field retrieval. Were there cloud targets with non-negligible U values? If so, how does that contribute to the interpretation of the retrieval results?

→ *For each individual pixel, the Stokes vector is rotated from the original measurement plane into the corresponding scattering plane (as in, e.g., 3.2.1 in Eshelman and Shaw, 2019). Each pixel has its own scattering plane. We think that our initial description was a bit misleading, and gave the impression, that we rotate all pixels into one plane which is the same for all pixels. In this case, only a narrow line of pixels has U ≈ 0, but this is not the case if the Stokes vectors are rotated into their own/unique scattering planes. We added the reference to Eshelman and Shaw, 2019, which is another description of the procedure and changed the sentence to:*

*"For further analysis, each measured Stokes vector is rotated into its pixel unique scattering plane (Hansen and Travis, 1974; Eshelman and Shaw, 2019) and we only evaluate Q."*

*Still, U can be non-zero in the scattering plane if the clouds are very inhomogeneous and asymmetrically distributed to the left and right side of the scattering plane. This is for example shown in Fig. 11 in Emde et al. (2018) based on model simulations. We verified that the U values are negligible for our case studies and that Q is much larger than U.*

*Eshelman, L. M. and Shaw, J. A.: Visualization of all-sky polarization images referenced in the instrument, scattering, and solar principal planes, Optical Engineering, 58, 082 418, https://doi.org/10.1117/1.OE.58.8.082418, 2019.*

*Emde C, Barlakas V, Cornet C, Evans F, Wang Z, Labonotte LC, et al. IPRT Polarized radiative transfer model intercomparison project three-dimensional test cases (phase b). J Quant Spectrosc Radiat Transfer 2018;209:19–44*

- Figure 2 caption: Please reword for clarity - "The primary bow of the cloudbow is visible in the degree of linear polarization as a bright ring at a scattering angle of about 140°"

→ *Changed as suggested.*

**Section 3**

- Line 173: "This method can easily be applied to any cloudbow observations, including those from commercial cameras, but it requires averaging over a large area." Please describe why is this less desirable than the co-located, along-track method presented in this paper.

→ *We changed the text to:*

*"An average cloudbow signal could be extracted from a cross-section of a single measurement (e.g., from Fig. 2). This method can easily be applied to any cloudbow observations, including those from commercial cameras, but the signal comes from a large area. The method presented in this paper is based on co-located observations along the track, which allows the acquisition of the cloudbow signature of individual targets. As a result, distributions are obtained at a high spatial resolution because this method does not involve averaging over a large area."*

- Line 175: Please change "we fly over it" to "specMACS images the scene".

  → *Changed as suggested.*

- Line 181: "In a final step, a look-up table (LUT) of cloudbow signals for different cloud droplet size distributions.." Please be specific that a simulated Mie LUT is used for this comparison.

  → *Changed as suggested.*

- Figure 3: The alpha angle in the figure is missing a zenith line coming directly from the plane, please add this in.

  → *Changed as suggested.*

- Figure 4: The colorbar is very large and doesn't apply to (d), so it would be cleaner visually if it was condensed and placed under (c). The caption would need to reflect this change as well. Please adapt Figure 10 similarly.

  → *Changed as suggested.*

- Line 240: Please remove the (single scattering). This is redundant with line 239.

  → *Changed as suggested.*

- Line 241: The polarized phase function is directly proportional to the measured polarized radiance Q, under the assumption of single scattering. P12 is not an approximation of Q. Please revise.

  → *We changed the sentence to:*

  *"The P 12 element is also called the polarized phase function and is approximately proportional to the measured polarized radiance Q in the scattering plane (Bréon and Goloub, 1998)."*

- Line 245: Effective variance determines the amplitude of the secondary maxima/minima (see Figure 5b), which is where the information content lies (not number of peaks). Please revise this sentence.

  → *We changed the sentence to:*

  *"The veff, however, determines the amplitude and widths of secondary minima of the radiance distribution but has only a small effect on the position of the minima."*

- Line 254: Please add the word unimodal or monomodal before "gamma distribution".

  → *Changed as suggested.*

- Line 280 and 287: Thanks for changing Eq. (6) and (8) to the typical/original convection!

  → *Changed as suggested.*

- Line 284: It has also been shown in Alexandrov et al. (2012a) that increasing cloud and/or aerosol optical thickness can impart a linear slope on Q. Reidi et al. (2010) shows that cirrus ice polarization signal is approximately linear in the rainbow region, too. These contaminations can also be accounted for by B and C terms in Eq. (6).

Riedi, J., Marchant, B., Platnick, S., Baum, B. A., Thieuleux, F., Oudard, C., Parol, F., Nicolas, J.-M., and Dubuisson, P.: Cloud thermodynamic phase inferred from merged POLDER and MODIS data, Atmos. Chem. Phys., 10, 11851–11865, https://doi.org/10.5194/acp-10-11851-2010, 2010.

> → *We changed the text to the following:*
>
> *"The fitting parameters B and C account for any remaining effects that are not considered in the single scattering assumption. For example, these could be contributions by multiple scattering. The term cos²(θ) corrects for Rayleigh scattering contributions (Alexandrov et al., 2012a). Other studies do not rely on the cosine term, and instead use a correction term linear in θ plus a constant (e.g., Bréon and Goloub (1998); Bréon and Doutriaux-Boucher (2005)). In the cloudbow range, however, this is similar to cos 2 (θ) (Alexandrov et al., 2012a). A further contribution beyond single-scattering could be a cirrus cloud above the cloud that generates the cloudbow. In Riedi et al. (2010) it was shown that the polarization signal of ice particles depends linearly on the scattering angle in the rainbow region. Furthermore, Alexandrov et al. (2012a) showed that the magnitude of a cloudbow signal is attenuated by an overlying aerosol layer, but the aerosol layer does not change the structure of the cloudbow signal. The fitting parameters B and C also account for these two effects of cirrus and aerosols."*

- Figure 6: The presentation of the "number of measurements" is a little confusing. In the plots, it almost looks like an uncertainty but it's a bit more complex than that and it blurs the overall message in my view. When the slope of the Q_fit signal is large in a small scattering angle range (see 6a between 142-147 degrees), it is hard to differentiate the box values from the measurement/fit lines. I recommend to replace the "number of measurements" pixels in the plots with errorbars that correspond to the angular measurement. This goes for Figures 9 and 15 as well.

> → *Changed as suggested. In addition, we added the corresponding RMSE and Qual values to the plot.*

**Section 4**

- Line 376: Please remove the comma between "example, to"

> → *Changed as suggested.*

- Line 378: Please remove the e.g. and commas.

> → *Changed as suggested.*

- Line 380: Please change to "In the case of small cumulus clouds, a precise geolocalization is important for image-to-image tracking."

> → *Changed as suggested.*

- Line 393: Please define 100m as a "target unit" here. This is definition referenced later in Line 401, but in an indirect way and was confusing on a blind read.

> → *We added the following text:*
>
> *"In the following, we will refer to this size of 100 m as "target unit".*

- Figure 10: See comment for Figure 4.

  *→ Changed as suggested.*

- Figure 10c: Why does the interpolation at the bottom right look artificial? Since it is not discussed in the text, I recommend to screen out this data and leave it unassessed (i.e. as it looks in 10a).

  *→ A single outlier with cloud top height = 5200 m was inside the stereo points dataset at the bottom right. The interpolation to all pixels of the image (panel c) is based on a linear interpolation of the stereo points, followed by a nearest-neighbour interpolation. The outlier was at the very edge of the image, where only few other stereo pixels were identified. Due to the nearest-neighbour interpolation, this artifact appeared. We removed the outlier from the dataset.*

- Line 423: If the error due to incorrect geolocalization is yet be estimated quantitatively, how is it that the cloudbow retrieval is "affected [by it] to a much lesser degree"? This is confusing. Please elaborate.

  *→ We explained it in more detail:*

  *"For a successful cloudbow retrieval, we rely on an accurate aggregation of the measurements by mapping from the known viewing angles to the image pixel location corresponding to the same cloud target. The stereographic approach of tracking cloud targets from one image frame to the next one provides exactly this information. It determines the cloud height by finding the ideal match between the known viewing directions during the overpass and the connecting line between identified targets at cloud top and the aircraft based on a matching contrast in different images. Some remaining residuum / mispointing error in the tracking, that stems from error sources in the geometric calibration (and as a result in cloud top height and horizontal wind), has negligible effects on the tracking. We manually verified the tracking of cloud targets with distinctive features during the overflight. One such example is shown in Fig. 11. Based on the location of the targets and the ambient wind at 18:29:40 UTC (panel b), the pixel positions of the targets in a previous (panel a) and a later image (panel c) are calculated. A visual comparison of the identified targets in the different images shows that the targets are successfully tracked, as the colored markers in Fig. 11 highlight the same areas of the cloud in all three images. Due to camera distortions the shape of the originally rectangular cloud targets (at 18:29:40 UTC) increasingly takes the shape of a trapezoid when they approach the edge of the entire image. Each panel in Fig. 11 shows only a small part of the entire image."*

- Line 433:

  ○ It is clear in 12b that the majority of 0.3+ veff values are "tracing" the lower boundary of the overlap between polA and polB cameras.

    *→ This is not the case, because the measurement comes from the polB camera and the overlap region lies in the upper part of the polB image (see Fig. 2). We added the information, that the measurements come from the polB camera to the text. We also made clear in Chapter 3.2, that the presented results are based on measurements of a single camera and that we do not (yet) combine the measurements of the two cameras.*

- 12f shows a big spike at the 0.32 veff bin, which is the very edge of the Mie LUT described on Line 269. This together suggests that these retrievals are artificially converging (either due to noise or errors in geolocalization) and may not be valid, even if they pass the Qual and RMSE. I suggest removing this veff bin from the Figure 7f and 12f plots (and corresponding data points from the 7b and 12b maps) to focus the discussion on the valid larger veff that do exist in the data.

  → *We again checked the targets with large veff. We think that there are some targets that show such large veff values which is why we do not want to rigorously filter out all veff > 0.32. The following argumentation is copied from the authors' response to reviewer #1.*

  - *Some signals actually came from ocean targets. In this case, the structure of the aggregated signal is relatively linear and the fit function can be fitted perfectly to the signal by keeping the parameter A (which scales the phase function) in the fitting function small. The fit has a very small RMSE, and as Qual ~ 1/RMSE this might also lead to a relatively high Qual index (at least > 2 as our original Qual filter threshold) and the signal is not filtered out.*

  - *Other signals (especially at cloud edges) did probably suffer from errors in the aggregation of angles. These did show a cloudbow signature, but the signal was not very clear, which could result from a mixture with ocean measurements. We increased the Qual filter threshold from 2 to 4. This did also filter out some targets, that are located quite central within the clouds, but where the cloud showed a lot of variability (shadows) and where an error in the aggregation of angles also could have a large effect.*

  - *There were some targets with veff = 0.32 which were located quite central within clouds (especially in case study 1) where the cloud did not show a lot of variability. The signals looked good and we think that these targets do have a veff of 0.32 or even larger. Such targets were not filtered out by the new Qual threshold of 4 and we decided to keep them in our results.*

- Line 458: McBride et al. (2020) was also careful to note that the subpixel reff and veff distribution that contributed to their larger-scale wide DSD result could also be impacted by cloud height georegistration (they used a granule-wide average, not pixel-by-pixel as in this study). I suggest to change the "does" to "may" in this sentence.

  → *Changed as suggested.*

**Section 5**

- Line 468: Remove the comma between "cloud, that" and change "are" to "may be". This has not been shown in this study, only suggested.

  → *Changed as suggested.*

- Line 470: Add a comma between "layer very"

  → *Changed as suggested.*